# Responsible product design to mitigate excessive gambling: A scoping review and z-curve analysis of replicability

**William H. B. McAuliffe**[1]*, **Timothy C. Edson**[1,2], **Eric R. Louderback**[1], **Alexander LaRaja**[1], **Debi A. LaPlante**[1,2]

**1** Division on Addiction Cambridge Health Alliance, A Harvard Medical School Teaching Hospital, Malden, MA, United States of America, **2** Harvard Medical School, Boston, MA, United States of America

* williamhbmcauliffe@gmail.com

## Abstract

### Objectives

Systematic mapping of evaluations of tools and interventions that are intended to mitigate risks for gambling harm.

### Design

Scoping Review and z-curve analysis (which estimates the average replicability of a body of literature).

### Search strategy

We searched 7 databases. We also examined reference lists of included studies, as well as papers that cited included studies. Included studies described a quantitative empirical assessment of a game-based (i.e., intrinsic to a specific gambling product) structural feature, user-directed tool, or regulatory initiative to promote responsible gambling. At least two research assistants independently performed screening and extracted study characteristics (e.g., study design and sample size). One author extracted statistics for the z-curve analysis.

### Results

86 studies met inclusion criteria. No tools or interventions had unambiguous evidence of efficacy, but some show promise, such as within-session breaks in play. Pre-registration of research hypotheses, methods, and analytic plans was absent until 2019, reflecting a recent embrace of open science practices. Published studies also inconsistently reported effect sizes and power analyses. The results of z-curve provide some evidence of publication bias, and suggest that the replicability of the responsible product design literature is uncertain but could be low.

**Data Availability Statement:** All data, including the "minimal dataset," and syntax can be found here: https://osf.io/m3nju/.

**Funding:** This research will be supported primarily by a research contract between the Division on Addiction and GVC Holdings PLC (hereafter, GVC). GVC is a large international gambling and online gambling operator. GVC had no involvement with the development of our research questions or protocol. They will not see any associated materials (i.e., retrieved studies, charted data, and manuscripts in preparation) while the study is in progress or have any editorial rights to any resulting manuscripts. GVC communication about this work will require approval of the Division on Addiction. GVC is now called Entain (https://entaingroup.com/).

**Competing interests:** The Division Current Funders: The Division on Addiction currently receives funding from the Addiction Treatment Center of New England via SAMHSA; EPIC Risk Management; The Foundation for Advancing Alcohol Responsibility (FAAR); DraftKings; the Gavin Foundation via the Substance Abuse and Mental Health Services Administration (SAMHSA); GVC Holdings, PLC; The Healing Lodge of the Seven Nations via the National Institutes of Health (National Institute of General Medical Sciences and National Institute on Drug Abuse); Health Resources in Action via the Massachusetts Department of Public Health Office of Problem Gambling Services; The Integrated Centre on Addiction Prevention and Treatment of the Tung Wah Group of Hospitals, Hong Kong; St. Francis House via the Massachusetts Department of Public Health Bureau of Substance Addiction Services; and the University of Nevada, Las Vegas via MGM Resorts International. Division Funders within the Last Five Years: The Division on Addiction currently receives funding from the Addiction Treatment Center of New England via SAMHSA; EPIC Risk Management; The Foundation for Advancing Alcohol Responsibility (FAAR); the Gavin Foundation via the Substance Abuse and Mental Health Services Administration (SAMHSA); GVC Holdings, PLC; The Healing Lodge of the Seven Nations via the National Institutes of Health (National Institute of General Medical Sciences and National Institute on Drug Abuse); Health Resources in Action via the Massachusetts Department of Public Health Office of Problem Gambling Services; The Integrated Centre on Addiction Prevention and Treatment of the Tung Wah Group of Hospitals, Hong Kong; St. Francis House via the Massachusetts Department of Public Health Bureau of Substance Addiction Services; and the University of Nevada, Las Vegas via MGM Resorts International. During the past 5 years, the Division on Addiction has also received funding from Aarhus University Hospital with funds

## Conclusion

Greater transparency and precision are paramount to improving the evidence base for responsible product design to mitigate gambling-related harm.

## Introduction

Interventions and tools for the safe use of inherently risky consumer products take many forms. For example, cars have mandatory *structural* features that autonomously mitigate the effects of accidents (e.g., airbags and crumple zones), include optional *user-directed* tools that assist with safe driving practices (e.g., turn signals and seat belts), and are subject to *regulations* that promote safe driving at large (e.g., minimum age of operator requirements and speed limits). Understanding the strength of evidence for various safety features and interventions can help stakeholders decide which to implement.

There is concern that popular gambling products, especially electronic gaming machines and internet gambling platforms, include features that increase risky gambling behavior [1, 2]. For instance, the ability to prematurely stop the reels of a video slot machine that has predetermined outcomes might give users an illusion of control over the outcome, motivating them to play longer [3]. Researchers have called for a greater emphasis on implementing safety features and interventions for gambling products [4]. However, it would be premature to make implementation recommendations without first determining whether existing evidence is based on sound research practices. Here, we report findings from a scoping review that quantitatively summarizes key features of existing research on game-based responsible gambling tools and interventions [5]. We identify trends in how studies are conducted, the state of knowledge about each type of tool, and whether a formal meta-analysis would be valuable. We also use the main result from each study to estimate the replicability of research on product safety in gambling.

### Product safety for gamblers

Responsible product design encompasses the obligatory efforts of government and industry actors to protect consumers, as well as the empowerment of gamblers to make informed decisions [6]. Structural features might facilitate this goal by reducing the likelihood and impact of an inadvertent period of excessive gambling. Tools only are structural if they operate beyond the user's control. For example, Rockloff and colleagues [7] introduced an automatic disqualification from jackpot wins after a fixed number of bets to disincentivize users from persisting in their play. Tools are user-directed if the user is not required to interact with them or can opt out of them. User-directed tools might empower gamblers by providing them with accurate information and concrete means of regulating their own temptations to gamble. For instance, pre-commitment systems for Electronic Gaming Machines (EGMs) allow users to voluntarily set limits for spending or losses, with mechanisms that interrupt play when reaching these limits [8]. Regulatory initiatives mandate the provision of accurate information and limit features that might facilitate excessive gambling. For example, Norway temporarily banned EGMs in 2007, later reintroducing new versions with less intense audio and visual stimuli, no banknote acceptors or payouts, and limits for daily and monthly losses [9].

Decisions to implement product designs intended to facilitate responsible gambling should be based on scientific assessment rather than intuitions or anecdotal evidence. Interventions

approved by The Danish Council for Independent
Research; ABMRF – The Foundation for Alcohol
Research; Caesars Enterprise Services, LLC; the
David H. Bor Library Fund, Cambridge Health
Alliance; DraftKings; Fenway Community Health
Center, Inc.; Massachusetts Department of Public
Health, Bureau of Substance Addiction Services;
Massachusetts Gaming Commission,
Commonwealth of Massachusetts; and University
of Nevada, Las Vegas via MGM Resorts
International. During the past five years, Debi A.
LaPlante has served as a paid grant reviewer for
the National Center for Responsible Gaming
(NCRG; now International Center for Responsible
Gaming), received travel funds, speaker honoraria,
and a scientific achievement award from the ICRG,
has received speaker honoraria and travel support
from the National Collegiate Athletic Association,
received honoraria funds for preparation of a book
chapter from Universite Laval, received publication
royalty fees from the American Psychological
Association, and received course royalty fees from
the Harvard Medical School Department of
Continuing Education. Dr. LaPlante is a non-paid
member of the New Hampshire Council for
Responsible Gambling. During the past 5 years,
Eric R. Louderback has received research funding
from a grant issued by the National Science
Foundation (NSF), a government agency based in
the United States. His research has been financially
supported by a Dean's Research Fellowship from
the University of Miami College of Arts & Sciences,
who also provided funds to present at academic
conferences. He has received travel support funds
from the Hebrew University of Jerusalem to
present research findings. William H. B. McAuliffe,
Timothy C. Edson and Alexander La Raja have no
funding disclosures to declare.

that have not been thoroughly vetted might have harmful effects that outweigh any intended benefits [10]. Furthermore, the implementation of tools that appear to have promise but are in fact ineffective could provide industry actors with unwarranted moral cover from advocates of more invasive interventions [11].

## Existing reviews of responsible product design for gambling

There have been several qualitative reviews of the empirical evidence for responsible gambling interventions as of 2015 (for an umbrella review, see [12]. These include reviews of structural features in electronic gambling [13], user-directed tools, such as self-exclusion [14], government and industry initiatives [15], product safety tools within real gambling environments [16], and EGM warning messages [17]. Although the reviews vary in their takeaway messages, they all stress that existing studies are limited by (a) relying on retrospective self-report, (b) using observational methods without incorporating features that address threats to causal inference, or (c) studying non-gamblers in laboratory settings. A more basic desideratum is whether published studies have yielded replicable findings. In addition to systematically charting the characteristics of available product safety evaluations, our review makes a unique contribution by focusing on replicability. Because opacity in how studies were conducted and analyzed undermines replicability, we also quantify the transparency of the responsible product design literature.

## Replicability of responsible product design research

Collaborative efforts to estimate the replicability of studies published in eminent journals, [18, 19] as well as the increasing number of individual replication attempts [20], have undermined confidence in numerous foundational findings in the social sciences. Central reasons for poor replicability include low statistical power [21], undisclosed methodological decisions that artificially inflate type I error rates (often called "questionable research practices" or "researcher degrees of freedom"), and publication bias abetted by incentives for novel, positive findings [22].

Researchers can use z-curve to estimate a literature's replicability [23]. Z-curve estimates the mean power of a set of studies with significant effects. Because we do not know which studies test true alternative hypotheses, "power" here does *not* refer to probability of obtaining a significant result conditional on the null hypothesis being false. Instead, power is the *unconditional* probability of obtaining a significant result, or "the percentage of significant results if the original studies were replicated exactly" (p.13). To our knowledge, researchers have not yet applied z-curve analyses to any segment of the gambling literature. Brunner and Schimmack [23] find that z-curve outperforms p-curve, p-uniform, and maximum likelihood estimation in estimating mean power of a set of studies selected for significance when there is heterogeneity in effect sizes (pgs. 12–13). We expect heterogeneity in effect sizes because different researchers are studying the effects of different types of interventions.

**Transparency of responsible product design research.** Because many replicability issues are due to a lack of transparency about analytic decisions, we also coded the transparency of each study along several dimensions. First, we employ a broad conceptualization of potential conflicts of interest by coding not only for funding sources, but also for the presence of a conflict of interest statement and a disclosure of funding sources from the past five years. Second, we code for whether the study was pre-registered and, if so, whether a link to the pre-registration is available in the manuscript. Pre-registration is the process by which researchers pre-empt questionable research practices by publicly documenting their hypotheses, methods, and analytic strategies prior to commencing a study [24]. Third, we code for whether the study

contains a power analysis and, if so, whether it was computed *a priori* or *post-hoc*. Conducting and reporting an *a priori* power analysis incentivizes researchers to conduct adequately powered studies, which in turn increases the likelihood that significant effects reflect true effects [25]. Fourth, we code for whether the study accompanies the test statistics with effect sizes and, if so, whether such effects were unstandardized or standardized. Effect sizes help readers understand whether authors' qualitative description of an effect's practical importance is consistent with its actual magnitude [26]. Effect sizes also can contain information about (a) replicability, because tests of larger effects more often yield significant results, and (b) fidelity of the research process, because honestly reported tests of non-trivial hypotheses typically yield medium-to-small effect sizes [27, 28].

## Materials and methods

We drafted our research protocol using the Preferred Reporting Items for Systematic reviews and Meta-Analyses extension for Scoping Reviews (PRISMA-ScR) [5]. The pre-registration for this project, as well as transparent changes to the pre-registration, are available on the Open Science Framework (osf.io/m3nju/files/).

### Study inclusion criteria

We included studies if they (a) were peer-reviewed, (b) were published at any point prior to our search, (c) were written in English, and (d) describe a quantitative empirical assessment of a game-based (i.e., intrinsic to a specific gambling product, rather than restrictions or tools that are intended to reduce gambling harm across gambling products) structural feature, user-directed tool, or regulatory initiative to promote responsible gambling. We specified the first three criteria during our sample acquisition and the fourth criterion during our title and abstract inspection and full-text inspection.

### Information sources and search strategy

A PRISMA diagram (see Fig 1) displays a summary of our search for studies that meet the inclusion criteria. To identify potentially relevant studies, on February 5, 2020, we searched the following bibliographic databases covering a variety of scientific disciplines: Medline, Embase (medicine); PsycARTICLES, PsycINFO (psychology); Global Health (public health); the Education Resources Information Center [ERIC] (education); and the Social Science Premium Collection.

We searched abstracts for the following keywords: gambl*, betting, wager*, responsib*, regulat*, protect*, warn*, structural, and product safety. We used the following search combinations: (gambl* OR betting OR wager*) AND (responsib* OR regulat* OR protect* OR warn* OR structural OR "product safety"). Once we specified our initial sample of studies by employing the first three inclusion criteria during a database search, we eliminated duplicates resulting from databases containing overlap in their results. Then, three research assistants screened the titles and abstracts of 10% of all non-duplicates to assess whether they described an empirical test of a game-based intervention (Krippendorff's alpha = .77). When it was not clear from the title and abstract alone whether a paper met inclusion criteria, we retained the full text for inspection. Afterwards, research assistants resolved disagreements through discussion with the first author.

Next, research assistants divided the remaining retrieved studies into three groups and screened their titles and abstracts independently. After reading the full texts, the first author deemed 11 studies as irrelevant that research assistants had flagged as meeting inclusion criteria. Our analytic sample consisted of all of the eligible studies from the database search

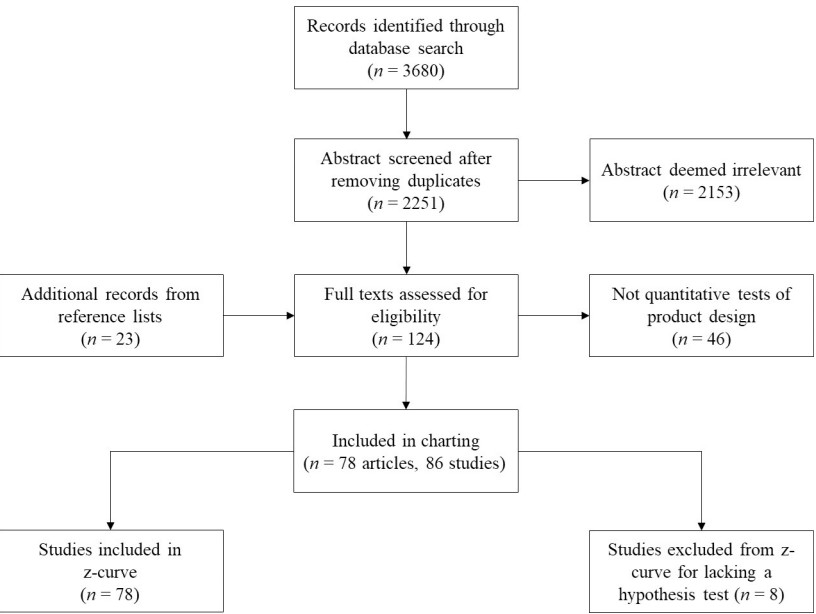

**Fig 1. PRISMA diagram of study selection and coding process.**

($N$ = 43), as well as ($N$ = 23) studies that the first author found between February 2020 and May 2020 after examining the reference lists of previous reviews, the studies that met inclusion criteria, and studies that cited the included studies according to Google Scholar. As a final quality check, the first author examined the abstracts that the research assistants indicated were irrelevant and found 12 that in fact met inclusion criteria. The final sample consisted of 78 journal articles with 86 relevant studies. See S1 Table.

## Data charting process

Raters charted the studies on the data items listed in Table 1 using Google Forms. We used an iterative process to determine the reliability of our charting. Two raters independently charted the data from a randomly selected subset of articles representing 10% of our eligible studies from the database search. Because the two coders' interrater reliability did not meet our standard (i.e., Cohen's **κ** is at least 0.70 and percentage agreement is at least 80% [29] after four iterations, we amended our pre-registration such that both raters charted all studies. The basis on which the first author resolved each discrepancy is available at https://osf.io/fa539/.

## Data charting process for study funder

We used funding sources' websites to code whether studies had ties to the gambling industry [30]. We counted a study as having direct gambling industry funding if any of the listed funding sources were directly part of the gambling industry (e.g., private companies such as Aristocrat Leisure Industries, and nationalized companies such as Loto-Q) or were non-profits funded by the gambling industry (e.g., International Center for Responsible Gaming). Non-industry funders included government agencies, universities, and private foundations. The websites on which we based our categorization decisions are available at https://osf.io/6sg9d/.

**Table 1. Data items, response format and response options.**

| Data item | Response format | Response options, if applicable |
|---|---|---|
| Study ID | Fill-in-the-blank | e.g., Nelson, S. E., LaPlante, D. A., Peller, A. J., Schumann, A., LaBrie, R. A., & Shaffer, H. J. (2008). Real limits in the virtual world: Self-limiting behavior of Internet gamblers. Journal of gambling Studies, 24(4), 463–477. |
| First Author Last Name | Fill-in-the-blank | e.g., Nelson |
| Study Funder | Fill-in-the-blank | e.g., International Center for Responsible Gaming |
| General Past 5-year Author Funding Statement | Select one | Yes, no |
| Conflict of Interest Statement | Select one | Conflicts reported, No Conflicts reported, No Conflict of Interest Statement |
| Month and year of earliest study publication | Fill-in-the-blank | e.g., March, 2016 Online First |
| Power analysis reported | Select one | Yes, *a priori*, Yes, *post-hoc*, Yes, timing unclear, No |
| Sample size (final analytic sample size reported after missing data is accounted for) | Fill-in-the-blank | e.g., N = 300 |
| Specific sample description | Fill-in-the-blank | e.g., "gambling treatment clients recruited from 2 treatment sites" |
| Study design | Gated question, Select one | (1) Experimental or Observational |
| | | (2a) Experimental: RCT, Non-randomized trial, or other |
| | | (2b) Observational: Cross-sectional, Prospective Cohort, Retrospective Cohort, or Case Series/Case Study, or other |
| Gambling concept(s) measures | Select all that apply / Fill-in-the-blank | Gambling participation / involvement (e.g., frequency, money spent, type of gambling activity, age of onset of gambling) regardless of gambling problem, Presence / severity of problem gambling (includes gambling-related consequences, age of onset of gambling problem), and Other (specify) |
| Game-based tool or intervention type | Gated question, Select all that apply / Fill-in-the-blank | (1) User-directed tool, Structural feature tool, and/or Regulatory tool to promote responsible gambling, Other (specify) |
| | | (2) Regulatory tool: Game-based tool or Other regulation (specify) |
| Duration of follow up phase, if applicable | Fill-in-the-blank | e.g., 3 months |
| Source of gambling concept(s) measured | Select all that apply | Self-report, Proxy report, Gambling records, Financial records, Other (specify) |
| Did the authors statistically test the association between the game-based tool or intervention type and a gambling outcome? | Select one | Yes, no (i.e., descriptive only) |
| Did the results of the statistical test include an effect size measure? | Select one | Yes, standardized and unstandardized; Yes, standardized; Yes, unstandardized; No |
| Registration status | Select one | Pre-registration available, Pre-registration and registered report available, Pre-registration not available, No Pre-registration |
| Major finding(s) | Fill-in-the-blank (narrative summary; 2–3 sentences) | |
| Notes | Fill-in-the-blank (optional) | |

## Analytic strategy and synthesis of evidence

We examined separate cross-tabulations of intervention type (i.e., structural feature, user-directed tool, or regulatory initiative) with study funder, study design, and registration status. We also provided a narrative summary of the major findings related to the effectiveness of identified safety characteristics, organized by intervention type. Finally, we calculated a year-by-year summary count of the number of publications by intervention type. See https://osf.io/76wm4/ for the syntax to conduct these analyses.

## Z-curve

We used the *z-curve* package in R to conduct z-curve analyses [31]. Z-curve is based on the idea that a distribution of z-scores can be derived from the average power of an entire set of studies. That distribution is truncated at the critical *z*-value (typically 1.96) after selection for statistical significance. Z-curve takes as input the set of significant findings (to mimic the editorial process of publishing only positive findings) and uses this truncated distribution to estimate the most likely shape of the non-truncated distribution of the population represented by the significant studies. To account for heterogeneity in effect sizes and power, z-curve estimates the distribution of all conducted studies using a finite mixture model of seven distributions, centered on z-scores of 0,1, 2, 3, 4, 5, and 6, respectively. An expectation maximization algorithm is used to assign studies probabilities of belonging to each distribution [32].

The resulting estimate of the non-truncated *z*-score distribution enables the computation of several statistics. First, the area under the curve to the right of the significance criterion is the Estimated Discovery Rate, or the estimated proportion of all studies that have been conducted that had significant results. The Observed Discovery Rate represents the proportion of coded tests that had significant results in the hypothesized direction. Because our dataset represents the entire population of interest, we omit confidence intervals from our reports of the Observed Discovery Rate. Evidence for publication bias exists if the Observed Discovery Rate is higher than the upper confidence limit of the Estimated Discovery Rate.

The Estimated Discovery Rate can be used to estimate how many non-significant results there might be for each significant result. This "file-drawer ratio" is equal to the estimated proportion of non-significant results (1- Estimated Discovery Rate) divided by the Estimated Discovery Rate. The file-drawer ratio can in turn be used to compute the False Discovery Risk, or the maximum proportion of significant studies that could represent false positives. The False Discovery Risk equals the product of the file-drawer ratio and the ratio of alpha (viz., .05) to 1-alpha.

Finally, the Expected Replication Rate is the mean power of the non-truncated distribution, and represents the estimated proportion of significant studies that would yield another significant effect if subjected to a direct replication. However, commentators frequently point to differences between original studies and replication studies that could explain why the former yield larger effect sizes than the latter [33, 34]. Consequently, the Expected Discovery Rate should more accurately predict the outcome of actual replication efforts than the Expected Replication Rate.

We included all 78 studies that contained inferential tests of their key hypothesis in the z-curve analysis. Two articles used the same dataset; we included only the first publication in z-curve, as the second article examined moderators of the findings reported in the first article.

If studies did not report exact *p*-values for significant effects, we computed them based on the sample size and either the descriptive statistics or test statistic using either base packages functions (e.g., the *pf* function for the *F* distribution) or the *compute.es* package in R [35]. We contacted authors for this information when they did not include the minimally sufficient

information in the paper. Two studies reported at least one *p*-value as less than .001, and we were unable to manually compute an exact value. After failing to receive clarification from the original authors, we treated these *p*-values as .0009 in the analysis.

We assigned studies with non-significant results a *p*-value of .300 if they did not report exact value and we could not reconstruct the exact *p*-value from the results reported in the paper (*n* = 8). The value of non-significant results does not impact the outcome of z-curve. We also computed exact *p*-values for studies that reported *p*-values to fewer than three decimal places. We recorded the exact *p*-values of significant effects in the opposite direction of what was hypothesized (*n* = 3) but treated them as non-significant (arbitrarily assigning them *p* = .300 for the purposes of the z-curve analysis). In a sensitivity analysis, we excluded studies that used a significance criterion other than a two-tailed alpha of .05 to evaluate the *p*-value of interest (*n* = 8), as z-curve's model of publication bias is based on censoring *z*-scores smaller than 1.96.

The *zcurve* function assumes that it is possible to identify a single key hypothesis test in each study. We anticipated that many studies in our review would regard multiple hypotheses as of equal importance or report multiple tests of the same hypothesis (e.g., using slightly different measures to represent the same dependent variable). We used the following strategies to select the "most focal hypothesis test": (a) In cases where authors tested the effect of interventions of varying dosage, we treated the test of the strongest intervention vs. the control condition as the most focal hypothesis test (e.g., if the effect of a short break and the effect of a long break are each compared to a no-break control condition, we would regard the comparison between taking a long break and taking no break as most focal); (b) When there were multiple dependent variables that were equally relevant to the central hypothesis, we randomly chose which test to regard as most focal; (c) When not all hypotheses were relevant to promoting responsible gambling, we only treated the hypotheses relevant to responsible gambling as candidates for the most focal hypothesis.

We also conducted a sensitivity analysis in which we repeated the analysis ten times, each time *randomly* selecting which test from each study to regard as focal by using a different seed number in *R*. After discovering that a small number of studies had a very large number of focal tests, we limited the number of potentially focal tests to six per study. To estimate upper- and lower-limits of replicability, we also re-reran z-curve once using the highest *p*-value from each study, and once more using the smallest *p*-value available from each study. The z-curve dataset is available at https://osf.io/acf3r/; the syntax we used to manually compute *p*-values and conduct z-curve is available at https://osf.io/aj6eu/.

## Results

### Characteristics of included studies

See Table 2 for the main characteristics of each study. We also created a Characteristics of Included Studies table that fully summarizes each study in terms of the charted items (see https://osf.io/k9sbq/).

### Study design

Of the 86 included studies, 97.7% (*n* = 84) of studies included at least one statistical test of the association between the game-based tool or intervention type and a gambling outcome. We observed that 69.8% (n = 60) of all studies were experimental, 91.7% (n = 55) of which randomly assigned participants to condition (i.e. 'true' experiments). The other 5 studies were quasi-experimental in that they contained multiple conditions but did not randomly assign participants to conditions. Also, 30.2% (*n* = 26) of all studies were observational. Among these

**Table 2. Characteristics of studies included in scoping review.**

| Reference | Study | Intervention | Study Design | Sample Size |
|---|---|---|---|---|
| Armstrong, T., et al. (2018). Exploring the effectiveness of an intelligent messages framework for developing warning messages to reduce gambling intensity. *Journal of Gambling Issues* 38: 67–84. [36] | 1 | Structural feature tool | Experimental | 172 |
| Armstrong, T., et al. (2019). Encouraging gamblers to think critically using generalised analytical priming is ineffective at reducing gambling biases. *Journal of Gambling Studies*, 36, 851–869. [37] | 1 | Structural feature tool | Experimental | 178 |
| Auer, M. & M. D. Griffiths (2013). Voluntary limit setting and player choice in most intense online gamblers: An empirical study of gambling behaviour. *Journal of Gambling Studies*, 29 (4): 647–660. [38] | 1 | User-directed tool | Observational | 5000 |
| Auer, M. M. & M. D. Griffiths (2015). Testing normative and self-appraisal feedback in an online slot-machine pop-up in a real-world setting." *Frontiers in Psychology*, 6. [39] | 1 | Structural feature tool | Observational | 23110 |
| Auer, M. M. & M. D. Griffiths (2015). The use of personalized behavioral feedback for online gamblers: An empirical study. *Frontiers in Psychology*, 6. [40] | 1 | User-directed tool | Observational | 1015 |
| Auer, M. M. & M. D. Griffiths (2016). Personalized behavioral feedback for online gamblers: A real world empirical study." *Frontiers in Psychology*, 7. [41] | 1 | Structural feature tool | Experimental | 5528 |
| Auer, M., & Griffiths, M. D. (2020). The use of personalized messages on wagering behavior of Swedish online gamblers: An empirical study. *Computers in Human Behavior*, 106402 [42] | 1 | Structural feature tool | Observational | 7134 |
| Auer, M., et al. (2014). Is "pop-up" messaging in online slot machine gambling effective as a responsible gambling strategy? *Journal of Gambling Issues*, 29: 1–10. [43] | 1 | Structural feature tool | Observational | 200000 |
| Auer, M., et al. (2018). The effect of loss-limit reminders on gambling behavior: A real-world study of Norwegian gamblers. *Journal of Behavioral Addictions*, 7(4): 1056–1067. [44] | 1 | Structural feature tool | Observational | 9384 |
| Auer, M., et al. (2019). An empirical study of the effect of voluntary limit-setting on gamblers' loyalty using behavioural tracking data. *International Journal of Mental Health and Addiction*, 1–12. [45] | 1 | User-directed tool | Observational | 165622 |
| Auer, M., et al. (2019). The effects of a mandatory play break on subsequent gambling among Norwegian video lottery terminal players. *Journal of Behavioral Addictions*, 8(3): 522–529. [46] | 1 | Structural feature tool | Observational | 1331 |
| Auer, M., et al. (2019). The effects of voluntary deposit limit-setting on long-term online gambling expenditure. *Cyberpsychology, behavior and social networking*, 23(2), 113–118. [47] | 1 | User-directed tool | Observational | 49560 |
| Benhsain, K., et al. (2004). Awareness of independence of events and erroneous perceptions while gambling. *Addictive Behaviors* 29(2): 399–404. [48] | 1 | Structural feature tool | Experimental | 31 |
| Beresford, K., & Blaszczynski, A. (2020). Return-to-player percentage in gaming machines: Impact of informative materials on player understanding. *Journal of Gambling Studies*, 36(1), 51–67. [49] | 1 | User-directed tool | Experimental | 112 |
| Blaszczynski, A., Cowley, E., Anthony, C., & Hinsley, K. (2016). Breaks in play: Do they achieve intended aims? *Journal of Gambling Studies*, 32(2), 789–800. [50] | 1 | Structural feature tool | Experimental | 141 |
| Blaszczynski, A., et al. (2005). Structural characteristics of electronic gaming machines and satisfaction of play among recreational and problem gamblers. *International Gambling Studies*, 5(2): 187–198. [51] | 1 | Structural feature tool | Quasi-Experimental | 95 |
| " | 2 | Structural feature tool | Quasi-Experimental | 305 |
| Blaszczynski, A., Gainsbury, S., & Karlov, L. (2014). Blue Gum gaming machine: An evaluation of responsible gambling features. *Journal of Gambling Studies*, 30(3), 697–712. [52] | 1 | User-directed tool | Observational | 299 |
| Brevers, D., Noel, X., Clark, L., Zyuzin, J., Justin Park, J., & Bechara, A. (2016). The impact of precommitment on risk-taking while gambling: A preliminary study. *Journal of Behavioral Addictions*, 5(1), 51–58. [53] | 1 | User-directed tool | Experimental | 60 |
| Broda A, LaPlante DA, Nelson SE, LaBrie RA, Bosworth LB, Shaffer HJ. 2008. Virtual harm reduction efforts for internet gambling: effects of deposit limits on actual internet sports gambling behaviour. Harm Reduct J. 5:1. [54] | 1 | Structural feature tool | Observational | 47000 |
| Byrne, C. A., & Russell, A. M. (2019). Making EGMs Accountable: Can an informative and dynamic interface help players self-regulate? *Journal of Gambling Studies*, 1–23. [55] | 1 | User-directed tool, Structural feature tool | Experimental | 213 |
| Caillon, J., et al. (2019). Effectiveness of at-risk gamblers' temporary self-Exclusion from Internet gambling sites. *Journal of Gambling Studies*, 35(2): 601–615. [56] | 1 | User-directed tool | Experimental | 60 |
| Choliz, M. (2010). Experimental analysis of the game in pathological gamblers: Effect of the immediacy of the reward in slot machines. *Journal of Gambling Studies*, 26(2): 249–256. [57] | 1 | Structural feature tool | Experimental | 10 |

(*Continued*)

**Table 2.** (*Continued*)

| Reference | Study | Intervention | Study Design | Sample Size |
|---|---|---|---|---|
| Cloutier, M. et al. (2006). Responsible gambling tools: Pop-up messages and pauses on video lottery terminals. *The Journal of Psychology*, 140(5): 434–438. [58] | 1 | Structural feature tool | Experimental | 40 |
| Corr, P. J. & Thompson, S. (2014). Pause for thought: response perseveration and personality in gambling. *Journal of Gambling Studies*, 30(4), 889–900. [59] | 1 | Structural feature tool | Experimental | 42 |
| Delfabbro, P. (2008). Evaluating the effectiveness of a limited reduction in electronic gaming machine availability on perceived gambling behaviour and objective expenditure. *International Gambling Studies*, 8(2): 151–165. [60] | 1 | Regulatory tool to promote responsible gambling | Observational | 594 |
| " | 2 | Regulatory tool to promote responsible gambling | Observational | 400 |
| du Preez, K. P., Landon, J., Bellringer, M., Garrett, N., & Abbott, M. (2016). The effects of pop-up harm minimisation messages on electronic gaming machine gambling behaviour in New Zealand. *Journal of Gambling Studies*, 32(4), 1115–1126. [61] | 1 | Structural feature tool | Observational | 460 |
| Floyd, K., et al. (2006). Use of warning messages to modify gambling beliefs and behavior in a laboratory investigation. *Psychology of Addictive Behaviors*, 20(1): 69–74. [62] | 1 | Structural feature tool | Experimental | 120 |
| Folkvord, F., et al. (2019). Experimental evidence on measures to protect consumers of online gambling services. *Journal of Behavioral Economics for Policy*, 3(1), 20–29. [63] | 1 | Structural feature tool, user-directed tool | Experimental | 522 |
| " | 2 | Structural feature tool, user-directed tool | Experimental | 5997 |
| Gainsbury, S. M., et al. (2015). Optimal content for warning messages to enhance consumer decision making and reduce problem gambling. *Journal of Business Research*, 68(10): 2093–2101. [64] | 1 | Structural feature tool | Quasi-Experimental | 667 |
| Gainsbury, S., et al. (2015). Determining optimal placement for pop-up messages: Evaluation of a live trial of dynamic warning messages for electronic gaming machines. *International Gambling Studies*, 15(1): 141–158. [65] | 1 | Structural feature tool | Quasi-Experimental | 667 |
| Gallagher, T., et al. (2011). Effects of a Video Lottery Terminal (VLT) Banner on Gambling: A Field Study. *International Journal of Mental Health and Addiction*, 9(1): 126–133. [66] | 1 | Structural feature tool | Observational | 54 |
| Ginley, M. K., et al. (2016). Gambling warning messages: The impact of winning and losing on message reception across a gambling session. *Psychology of Addictive Behaviors*, 30(8): 931–938. [17] | 1 | Structural feature tool | Experimental | 154 |
| Griffiths, M. D., et al. (2009). Social responsibility tools in online gambling: A survey of attitudes and behavior among Internet gamblers. *CyberPsychology & Behavior*, 12(4): 413–421. [67] | 1 | User-directed tool | Observational | 2348 |
| Hansen, M. B. & I. M., Rossow. (2012). Does a reduction in the overall amount of gambling imply a reduction at all levels of gambling? *Addiction Research & Theory*, 20(2): 145–152. [68] | 1 | Regulatory tool to promote responsible gambling | Observational | 27845 |
| Hansen, M., & Rossow, I. (2010). Limited cash flow on slot machines: Effects of prohibition of note acceptors on adolescent gambling behaviour. *International Journal of Mental Health and Addiction*, 8(1), 70–81. [69] | 1 | Regulatory tool to promote responsible gambling | Observational | 62481 |
| Harris, A., & Parke, A. (2016). The interaction of gambling outcome and gambling harm-minimisation strategies for electronic gambling: The efficacy of computer-generated self-appraisal messaging. *International Journal of Mental Health and Addiction*, 14(4), 597–617. [70] | 1 | Structural feature tool | Experimental | 30 |
| Hayer, T. & G. Meyer (2011). Internet self-exclusion: Characteristics of self-excluded gamblers and preliminary evidence for Its effectiveness. *International Journal of Mental Health and Addiction*, 9(3): 296–307. [71] | 1 | User-directed tool | Observational | 259 |
| Hollingshead, S. J., et al. (2019). Do you read me? Including personalized behavioral feedback in pop-up messages does not enhance limit adherence among gamblers. *Computers in Human Behavior*, 94: 122–130. [72] | 1 | Structural feature tool | Experimental | 131 |
| " | 2 | Structural feature tool | Experimental | 109 |
| Hollingshead, S. J., et al. (2019). When should players be taught to gamble responsibly? Timing of educational information upregulates responsible gambling intentions. *Addiction Research & Theory* 27(6): 507–514. [73] | 1 | User-directed tool | Experimental | 98 |
| Ivanova, E. N., Magnusson, K., & Carlbring, P. (2019). Deposit limit prompt in online gambling for reducing gambling intensity: a randomized controlled trial. *Frontiers in Psychology*, 10, 639. [11] | 1 | Structural feature tool, User-directed tool | Experimental | 4328 |

(*Continued*)

**Table 2.** (Continued)

| Reference | Study | Intervention | Study Design | Sample Size |
|---|---|---|---|---|
| Jardin, B. & E. Wulfert (2009). The use of messages in altering risky gambling behavior in college students: an experimental analogue study. *The American Journal on Addictions*, *18*(3): 243–247. [74] | 1 | Structural feature tool | Experimental | 104 |
| Jardin, B. F. & E. Wulfert (2012). The use of messages in altering risky gambling behavior in experienced gamblers. *Psychology of Addictive Behaviors*, *26*(1): 166–170. [75] | 1 | Structural feature tool | Experimental | 80 |
| Kim, H. S., et al. (2014). Limit your time, gamble responsibly: Setting a time limit (via pop-up message) on an electronic gaming machine reduces time on device. *International Gambling Studies*, *14*(2): 266–278. [76] | 1 | Structural feature tool, User-directed tool | Experimental | 43 |
| Ladouceur, R. & Sevigny, S. (2006). The impact of video lottery game speed on gamblers. *Journal of Gambling Issues*, 17. [77] | 1 | Structural feature tool | Experimental | 43 |
| Ladouceur, R. & Sevigny, S. (2009). Electronic gambling machines: Influence of a clock, a cash display, and a precommitment on gambling time. *Journal of Gambling Issues*, *23*: 31–41. [78] | 1 | User-directed tool | Observational | 38 |
| Ladouceur, R., & Sevigny, S. (2003). Interactive messages on video lottery terminals and persistence in gambling. *Gambling Research*, 15(1), 45. [79] | 1 | Structural feature tool | Experimental | 30 |
| Lavoie, R. V., & Main, K. J. (2019). When losing money and time feels good: The paradoxical role of flow in gambling. *Journal of Gambling Issues*, 41. [80] | 1 | Structural feature tool | Experimental | 229 |
| " | 2 | Structural feature tool | Experimental | 62 |
| Loba, P., Stewart, S. H., Klein, R. M., & Blackburn, J. R. (2001). Manipulations of the features of standard video lottery terminal (VLT) games: Effects in pathological and non-pathological gamblers. *Journal of Gambling Studies*, *17*(4), 297–320. [81] | 1 | Structural feature tool, User-directed tool | Experimental | 60 |
| Luquiens, A., et al. (2019). Self-exclusion among online poker gamblers: Effects on expenditure in time and money as compared to matched controls. *International Journal of Environmental Research and Public Gealth*, 16(22), 4399. [82] | 1 | User-directed tool | Observational | 9774 |
| Luquiens, A., et al. (2018). Description and assessment of trustability of motives for self-exclusion reported by online poker gamblers in a cohort using account-based gambling data. *BMJ open*, *8*(12). [83] | 1 | User-directed tool | Observational | 1996 |
| May, R. K., et al. (2005). Gambling-related irrational beliefs in the maintenance and modification of gambling behaviour. *International Gambling Studies*, 5(2), 155–167. [84] | 1 | Structural feature tool | Experimental | 114 |
| McGivern, P., et al. (2019). The impact of pop-up warning messages of losses on expenditure in a simulated game of online roulette: a pilot study. *BMC Public Health*, 19(1): 822. [85] | 1 | Structural feature tool | Experimental | 45 |
| Monaghan, S. and A. Blaszczynski (2010). Impact of mode of display and message content of responsible gambling signs for electronic gaming machines on regular gamblers. *Journal of Gambling Studies*, 26(1): 67–88. [86] | 1 | Structural feature tool | Experimental | 127 |
| " | 2 | Structural feature tool | Experimental | 124 |
| Monaghan, S., & Blaszczynski, A. (2007). Recall of electronic gaming machine signs: A static versus a dynamic mode of presentation. *Journal of Gambling Issues*, *20*, 253–268. [87] | 1 | User-directed tool | Experimental | 92 |
| Monaghan, S., Blaszczynski, A., & Nower, L. (2009). Do warning signs on electronic gaming machines influence irrational cognitions? *Psychological Reports*, *105*(1), 173–187. [88] | 1 | User-directed tool | Experimental | 93 |
| Muñoz, Y., et al. (2010). Using fear appeals in warning labels to promote responsible gambling among VLT players: The key role of depth of information processing. *Journal of Gambling Studies*, *26*(4): 593–609. [89] | 1 | Structural feature tool | Experimental | 258 |
| Muñoz, Y., et al. (2013). Graphic gambling warnings: How they affect emotions, cognitive responses and attitude change. *Journal of Gambling Studies*, 29(3): 507–524. [90] | 1 | Structural feature tool | Experimental | 103 |
| Nelson, S. E., LaPlante, D. A., Peller, A. J., Schumann, A., LaBrie, R. A., & Shaffer, H. J. (2008). Real limits in the virtual world: Self-limiting behavior of Internet gamblers. *Journal of Gambling Studies*, *24*(4), 463–477. [91] | 1 | User-directed tool | Observational | 47134 |
| Newall, P. W. S., et al. (2020). Equivalent gambling warning labels are perceived differently. *Addiction*, *115*(9), 1762–1767. [92] | 1 | User-directed tool | Experimental | 390 |
| " | 2 | User-directed tool | Experimental | 407 |
| Newall, P. W. S., Walasek, L., & Ludvig, E. A. (2020). Percentage and currency framing of house-edge gambling warning labels. *International Journal of Mental Health and Addiction*, 1–8. [93] | 1 | User-directed tool | Experimental | 845 |

(*Continued*)

**Table 2.** (Continued)

| Reference | Study | Intervention | Study Design | Sample Size |
|---|---|---|---|---|
| Parke, A., et al. (2019). Effect of within-session breaks in play on responsible gambling behaviour during sustained monetary losses. *Current Psychology*, 1–13. [94] | 1 | Structural feature tool | Experimental | 74 |
| Phillips, J. G. & J. Landon (2016). Dynamic changes in the use of online advice in response to task success or failure. *Behaviour & Information Technology* 35(10): 796–806. [95] | 1 | User-directed tool | Experimental | 21 |
| Phillips, J. G. & R. P. Ogeil (2010). Alcohol influences the use of decisional support. *Psychopharmacology*, 208(4): 603–611. [96] | 1 | User-directed tool | Experimental | 16 |
| Phillips, J. G., & Ogeil, R. P. (2007). Alcohol consumption and computer blackjack. *The Journal of General Psychology*, 134(3), 333–353. [97] | 1 | User-directed tool | Experimental | 20 |
| Phillips, J. G., Laughlin, A. L., Ogeil, R. P., & Blaszczynski, A. (2011). Effects of directional decisional support upon risk taking online. *The Ergonomics Open Journal*, 4(1). [98] | 1 | User-directed tool | Experimental | 24 |
| Rockloff, M. J., et al. (2015). Jackpot expiry: An experimental investigation of a new EGM player-protection feature. *Journal of Gambling Studies*, 31(4): 1505–1514. [7] | 1 | Structural feature tool | Experimental | 107 |
| Sharpe, L., et al. (2005). Structural changes to Electronic Gaming Machines as effective harm minimization strategies for non-problem and problem gamblers. *Journal of Gambling Studies*, 21(4): 503–520. [99] | 1 | Structural feature tool | Quasi-Experimental | 210 |
| Steenbergh, T. A., et al. (2004). Impact of warning and brief intervention messages on knowledge of gambling risk, irrational beliefs and behaviour. *International Gambling Studies*, 4(1): 3–16. [100] | 1 | Structural feature tool | Experimental | 101 |
| Stewart, M. J. & M. J. A. Wohl (2013). Pop-up messages, dissociation, and craving: How monetary limit reminders facilitate adherence in a session of slot machine gambling. *Psychology of Addictive Behaviors*, 27(1): 268–273. [101] | 1 | Structural feature tool | Experimental | 59 |
| Tabri, N., et al. (2019). A limit approaching pop-up message reduces gambling expenditures, except among players with a financially focused self-concept. *International Gambling Studies*, 19(2): 327–338. [102] | 1 | Structural feature tool | Experimental | 88 |
| Thompson, S. J., & Corr, P. J. (2013). A feedback-response pause normalises response perseveration deficits in pathological gamblers. *International Journal of Mental Health and Addiction*, 11(5), 601–610. [103] | 1 | Structural feature tool | Experimental | 81 |
| Walker, A. C., Stange, M., Dixon, M. J., Koehler, D. J., & Fugelsang, J. A. (2019). Graphical depiction of statistical information improves gambling-related judgments. *Journal of Gambling Studies*, 35(3), 945–968. [104] | 2 | User-directed tool | Experimental | 200 |
| Wohl, M. J. A., et al. (2014). Building it better: Applying human-computer interaction and persuasive system design principles to a monetary limit tool improves responsible gambling. *Computers in Human Behavior*, 37: 124–132. [105] | 2 | Structural feature tool | Experimental | 56 |
| Wohl, M. J. A., et al. (2017). How much have you won or lost? Personalized behavioral feedback about gambling expenditures regulates play. *Computers in Human Behavior*, 70: 437–445. [106] | 1 | Structural feature tool | Observational | 649 |
| Wohl, M. J., et al. (2010). Animation-based education as a gambling prevention tool: Correcting erroneous cognitions and reducing the frequency of exceeding limits among slots players. *Journal of Gambling Studies*, 26(3), 469–486. [107] | 1 | User-directed tool | Experimental | 242 |
| Wohl, M. J., et al. (2013). Facilitating responsible gambling: The relative effectiveness of education-based animation and monetary limit setting pop-up messages among electronic gaming machine players. *Journal of Gambling Studies*, 29(4): 703–717. [108] | 1 | Structural feature tool, User-directed tool | Experimental | 72 |
| Wohl, M. J., Santesso, D. L., & Harrigan, K. (2013). Reducing erroneous cognition and the frequency of exceeding limits among slots players: A short (3-minute) educational animation facilitates responsible gambling. *International Journal of Mental Health and Addiction*, 11(4), 409–423. [109] | 1 | User-directed tool | Experimental | 123 |
| " | 2 | User-directed tool | Observational | 24 |
| Wood, R. T. A. and M. J. A. Wohl (2015). Assessing the effectiveness of a responsible gambling behavioural feedback tool for reducing the gambling expenditure of at-risk players. *International Gambling Studies*, 15(2): 1–16. [110] | 1 | User-directed tool | Observational | 1558 |

observational studies, 34.6% (*n* = 9) were cross-sectional, 46.2% (*n* = 12) were retrospective cohort studies, 7.7% (*n* = 2) were prospective cohort studies, and 11.5% (*n* = 3) were case series.

## Sample

Sample sizes varied widely, ranging from 10 to 200,000. The median sample size was 136. Authors most commonly sampled people who gamble in their everyday lives (70.9%; $n = 61$). 68.9% of these studies recruited gamblers from a convenience population ($n = 42$)—that is, from a nearby casino, advertisement in a local newspaper, online panel pre-screened for gamblers, gamblers in an undergraduate psychology student pool, etc. The rest (31.1%; $n = 19$) sampled gamblers from a gambling platform or casino loyalty program.

Among the 29.4% ($n = 25$) of studies that did not exclusively sample gamblers, 48% ($n = 12$) sampled community members and 52% ($n = 13$) sampled university students. Across all studies, 8.1% ($n = 7$) of studies used screening tools during enrollment to sample at-risk gamblers, and 11.6% ($n = 10$) used screening tools during enrollment to exclude at-risk gamblers from participation.

## Gambling concepts

Of the 86 included studies, 94.2% ($n = 81$) of studies measured gambling participation. 73.3% ($n = 63$) of studies measured the presence or severity of gambling-related problems. Included in this count are measures of gambling-related problems specifically, as well as measures of constructs that might be symptomatic of them, such as impulsivity. One study measured recall of the content of gambling messages, which we did not count as a measure of either gambling participation or gambling-related problems.

## Measurement method

Many studies relied on multiple measurement methods. 76.7% ($n = 66$) of studies used at least one self-report measure. 61.6% ($n = 53$) of studies used gambling records to measure at least one construct. Two studies used proxy reports (in these cases, by trained observers), and one used financial records.

We conducted unplanned chi-square tests to explore whether the choice to measure gambling participation or gambling-related problems might have influenced the measurement method. There were significantly more studies of gambling-related problems that used self-reports than would be expected by chance, $\chi^2(1) = 49.10$, $p < .001$, $\varphi_c = .48$. There was a non-significant tendency for studies of gambling participation to not use self-report measures, $\chi^2(1) = 0.52$, $p = .469$, $\varphi_c = -.08$. The non-significance of this pattern of results held when using a Fisher's exact test [111] to account for expected cell counts with five or fewer cases. A significantly higher proportion of gambling participation studies used gambling records, $\chi^2(1) = 5.98$, $p = .014$, $\varphi_c = .19$, and we obtained the same result using a Fisher's exact test. Finally, fewer studies of gambling-related problems used gambling records than would be expected by chance, $\chi^2(1) = 4.70$, $p = .031$, $\varphi_c = -.16$.

## Follow-up period

Of the 86 included studies, 29.1% ($n = 25$) of studies included a "follow-up" component (i.e., at least one measurement occasion beyond the day on which the intervention was implemented). The median follow-up length among studies that had a follow-up component was 60 days (minimum = 1 day, maximum = 1 year).

## Current and past funding sources

Some articles reported multiple sources of funding. Of all 78 included articles, 26.9% ($n = 21$) had direct funding from the gambling industry, 10.4% ($n = 8$) had university funding, 3.8%

($n = 3$) had funding from a private foundation, 39.7% ($n = 31$) had funding from a government agency, 10.3% ($n = 5$) received no funding, and 25.6% ($n = 20$) did not provide enough information about funding source to code. Of government-funded articles, 25.8% ($n = 8$) were from an agency that is funded by revenue from gambling.

## Conflict of interest statement

Of the 78 articles, 20.5% ($n = 16$) reported conflicts of interest, 30.8% ($n = 24$) reported that they had no conflicts of interest, and 48.7% ($n = 38$) did not include a conflict of interest statement. Only two studies provided an explicit statement about all sources of funding that authors had received in the past five years.

## Pre-registration status

Of the 86 included studies, 93% ($n = 80$) of studies made no mention of a pre-registration. 5.8% ($n = 5$) of studies had a pre-registration that we were able to access. An additional study mentioned a pre-registration, but the hyperlink did not work. All six studies that mentioned a pre-registration were published in either 2019 or 2020.

## Power analysis

Of the 84 studies that included a statistical test, 7.1% of studies ($n = 6$) reported an *a priori* power analysis. Another 6.0% ($n = 5$) reported a *post hoc* power analysis. 87.2% ($n = 75$) of studies did not report a power analysis justifying sample size.

## Effect size

Of the 84 studies that included a statistical test of an intervention on gambling, 64.3% ($n = 54$) reported at least one effect size. Of those, 37.0% ($n = 20$) reported at least one unstandardized effect size, 77.8% ($n = 42$) reported at least one standardized effect size, and 9.3% ($n = 8$) reported at least one unstandardized effect and at least one standardized effect size. Because of the very large number of tests reported in several studies, many of which were not related to the evaluation of responsible product design per se, we abandoned our pre-registered plan to "report what percentage of the test statistics we transcribe are accompanied by an effect size."

## Unplanned coding of sampled populations

While charting studies, we noticed that most studies were conducted in a small number of countries. To follow up on these anecdotal observations, we charted the country or countries from which each study sampled. The most commonly sampled countries were Australia (22.1%, $n = 19$), Canada (20.9%; $n = 18$), and the United States (10.5%, $n = 9$). There were six studies (6.9%) that sampled several countries, usually all from Europe, but in some cases from multiple continents. There were eight studies (9.3%) where authors did not specify the country (ies) where the research was conducted. Though studies examining *causes* of excessive gambling in Asian, African, or South American countries exist [e.g., 112, 113], none of the included studies sampled these populations.

We also formed the impression that most studies did not discuss threats to generalizability based on participant characteristics. Consequently, we transcribed statements in each article's discussion section about potential limitations based on the sampled population. 15.1% of studies ($n = 13$) explicitly mentioned limitations based on country or cultural milieu (e.g., ethnic group, socioeconomic status, etc.). Many discussion sections that did not mention cultural constraints *did* discuss how university students might differ from the general population, how

sampling players who favor a certain game might have impacted results, or how results from low-risk gamblers might not extend to high-risk gamblers.

## Game-based tool or interaction type

Some studies ($n = 7$) tested multiple types of responsible product designs. Of the 86 studies, 61.6% of studies ($n = 53$) tested structural tools, 41.9% ($n = 36$) tested user-directed tools, and 4.7% tested ($n = 4$) regulations. We used chi-square tests of independence to examine whether there is a relationship between intervention type and study funder, study design, or registration status. We excluded studies that investigated multiple types of tools ($n = 7$) from this analysis.

Intervention type had a non-significant association with pre-registration status, $\chi^2(4) = 9.20$, $p = .056$, $\varphi_c = .24$. The test was significant in an exploratory follow-up using Fisher's exact test to account for low expected cell counts, $p = .027$. This potential effect was driven by all pre-registered studies testing user-directed tools.

Intervention type significantly varied by study design, $\chi^2(2) = 13.73$, $p = .001$, $\varphi_c = .42$. Standardized residuals were significant (greater than 2 [111]) for structural features and regulations, but not user-directed tools. Structural tools were tested more often via experiments ($n = 37$) than by observational methods ($n = 9$). User-directed tools were tested by observational methods ($n = 13$) almost as often as by experiments ($n = 16$). Regulations were tested only in observational studies.

Last, we found that intervention type did not vary by industry-funded research status, $\chi^2(2) = 1.69$, $p = .429$, $\varphi_c = .15$. The non-significance of this pattern remained when we ran an unplanned analysis that counted studies ($n = 9$) sponsored by government agencies that are funded by earmarked tax revenue from gambling operators (e.g., research funded by Gambling Research Exchange Ontario) as industry-funded research.

## Temporal trends

Included studies were published between 2001 and 2020. See Fig 2 for the frequency of studies on structural features, user-directed tools, and regulations by year (the total count is more than 86 because some studies examined more than one type of tool). Researchers have published about two to five evaluations of structural feature tools annually since 2003, with a spike at 13 studies in 2019. Studies of user-directed tools have continued at a steady rate from 2007 to the present, with a peak at 10 in 2019. Regulations were studied only between 2008 and 2011.

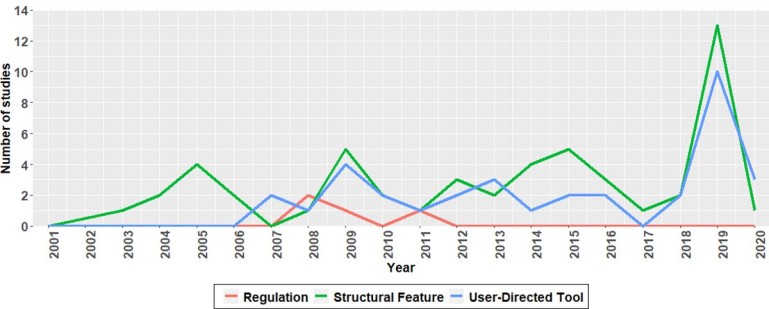

**Fig 2. Yearly counts of studies on structural features, user-directed tools, and regulations.** 2020 only includes studies through May.

### Narrative review

**Structural feature tools.**    Our charting suggested that the gambling research literature has examined three main structural feature tools: pop-up messages, breaks in play, and covert structural tools.

*Pop-up messages*. In all, 41 of 53 structural feature tool studies (77%) examined pop-up messages. Perhaps the most well-studied pop-up warning message seeks to educate participants of the statistical principles that explain why the expected value of gambling is negative. However, studies of so-called self-appraisal messages that encourage gamblers to reflect on whether their current gambling behavior is consistent with their goals are also common.

There were 20 pop-up message studies (49%) suggesting that pop-up messages had a favorable responsible gambling impact. For example, Jardin and Wulfert [74, 75] compared pop-ups that remind participants about the chance-based nature of gambling to pop-ups with trivia and a control condition with no pop-ups. Participants who received reminder messages lost less money and made fewer bets than those who read trivia or did not see a pop-up.

We observed that 7 pop-up message studies (17%) suggested pop-up messages had no responsible gambling impact. For example, Lavoie and Main [80] presented pop-ups just before playing slots which warned that gambling can produce a state of immersion that can cause excessive spending. In a second study, they presented pop-ups in the middle of blackjack to inform the user how long he or she had been playing. The pop-ups did not reduce immersion or time or money spent gambling in either study.

Finally, our charting indicated that 14 pop-up message studies (34%) suggested pop-up messages had mixed impact. For example, Tabri, Hollingshead, and Wohl [102] had participants set money limits, and varied whether participants (a) received a single warning message asking whether they would like to continue when the limit was reached, or (b) also received a message when they were close to reaching their limit. Messages about approaching limits increased stopping of play before participants reached their limits. This effect was strong (i.e., an odds ratio above 31) for participants who did not have a financially focused self-concept (i.e., those who do not define themselves in terms of financial success), but was non-significant for those with a financially focused self-concept (i.e., those who self-worth is tied up in financial success).

*Breaks in play*. In all, 7 of 53 structural feature tool studies (13%) examined breaks in play. The typical justification for mandatory pauses in between rounds or after playing for a certain duration is that gambling induces in at-risk players a dissociative state that undermines rational decision-making [114, 115]. In games of chance, the break in play would purportedly mitigate excessive gambling by lifting players out of their trance. In games of skill, a forced pause would give losing players the time to reevaluate their strategy.

There were 3 breaks in play studies (43%) suggesting favorable responsible gambling impact. For example, people with [103] and without gambling-related problems [59] from Wales played card games in which winning became less likely over time. Imposing a 5-second pause between bets reduced the number of rounds played and the magnitudes of monetary losses.

We observed that 3 breaks in play studies (43%) had no or unfavorable responsible gambling impact. For example, one experiment [50] varied whether Australian university students received no break, a 3-minute break, or an 8-minute break from blackjack. The results indicated that both 3-minute and 8-minute breaks increased cravings to gamble and did not decrease dissociative feelings.

Finally, one breaks in play study (14%) suggested mixed responsible gambling impact [94]. The authors found that a three-minute break did increase the response latency between rounds

for EGM players in the face of consistent losses. However, this slowed play did not translate into playing fewer trials.

*Covert structural tools.* In all, 6 of 53 structural feature tool studies (11%) examined covert structural tools. Covert interventions are intended to affect the proximate causes of excessive gambling without requiring buy-in from the gambler. One covert intervention study (17%) reported no responsible gambling impact [37]. The authors examined whether an implicit prime of analytic thinking, or a stimulus that is designed to induce a reflective mindset without the participant realizing that the stimulus has this effect, would attenuate erroneous gambling beliefs and gambling intensity in a sample of EGM players. Randomly assigning participants to unscramble sentences that either did or did not include words related to rationality had no significant effects on gambling beliefs or behavior.

Our charting indicated that 5 covert intervention studies (83%) yielded mixed responsible gambling impact. For example, Sharpe and colleagues [99] modified EGMs in Australian venues with varying levels of reel speeds, maximum bet restrictions, and restrictions on the maximum banknote accepted. Limiting the maximum bet to $1 reduced the number of bets and losses relative to EGMs with $10 bet maximums, but modifying note acceptors and reel speeds did not reduce gambling activity.

**User-directed tools.** Our charting suggested that the gambling research literature has examined two main user-directed feature tools: precommitment and information aids.

*Precommitment.* In all, 16 of 36 user-directed tool studies (44%) examined precommitment. Precommitment involves prospectively planning to restrict one's own ability to gamble excessively. Its efficacy is premised on players recognizing that they have difficulty exercising self-control "in the heat of the moment."

Our charting revealed 6 precommitment studies (38%) finding that precommitment had a favorable responsible gambling impact. Brevers and colleagues [53] presented participants with a gambling task with four options that varied in reward and risk. During "precommitment" trials, there was a preliminary step in which participants could remove the high-risk, high reward options from the trial. In the control trials, all four options were always available. Participants eliminated the high-risk options in about half of precommitment trials, resulting in lower-risk decisions in the precommitment condition.

We observed 4 precommitment studies (25%) suggesting that precommitment had no or unfavorable responsible gambling impact. In arguably the best-designed study that met inclusion criteria [11], researchers manipulated whether Finnish online gamblers were presented with a prompt to consider setting a deposit limit. Deposit limits control how much users can wager over a certain period. The prompts greatly increased limit-setting relative to a no-prompt control condition. But this increased limit-setting did not reduce net loss, total number of days gambled, or amount of money deposited in the following 90 days.

Finally, 6 precommitment studies (38%) reported that precommitment had mixed responsible gambling impact. To illustrate, Caillon and colleagues [56] randomly assigned users to self-exclude from an online gambling platform for a week. Self-exclusion prevents the user from using a certain gambling platform at all. Self-exclusion did not cause significant differences in money or time spent gambling fifteen days or two months after the self-exclusion began. There were, however, decreases in gambling illusions and desire to gamble two months later.

*Information aids.* In all, 24 of 36 user-directed tool studies (67%) examined information aids. Information aids provide the user with facts about a game's payout structure. They should reduce gambling to the extent that they correct misbeliefs that cause excessive play and should minimize losses to the extent that they help users make statistically optimal decisions.

There were 8 information aid studies (33%) suggesting that information aids had a favorable responsible gambling impact. For example, when scratch card players had access to both the number of unclaimed prizes left and the payback percentage (the ratio of unclaimed prizes to total scratch cards remaining) in a numerical format, participants were influenced by the former even though it is only the latter that is diagnostic [104]. A follow-up experiment presented the same payback percentage information using a visual (a five-star rating system). Participants now more often relied on the payback percentage to make decisions about scratch cards, even though they did not understand the concept of payback percentage any better than participants in the first experiment. The visual rating system might have been interpreted by participants as recommendations for or against a given scratch card. So long as participants trusted the testimony of the experimenters, there was no need for participants to understand why payback percentages are more germane than counts of unclaimed prizes.

We observed that 6 information aid studies (25%) suggested information aids had no or unfavorable responsible gambling impact. For example, Beresford and Blaszczynski [49] tested multiple formats to improve understanding of return-to-player percentage. The concept refers to the percentage of money that a game returns to players in the long run, but players often believe that it approximates the percentage of stakes that remain with the average player at the end of individual sessions. This belief is incorrect because EGM winnings are designed to vary substantially in the short-term, and reinvesting wins tends to reduce earnings to zero. The authors reported that neither an infographic, a vignette, or a brochure increased understanding of return-to-player percentage relative to the mandatory signage on EGMs in South Australia.

Finally, our charting indicated that 10 information aid studies (42%) reported mixed responsible gambling impact. For instance, "Basic strategy" is a decision tool for reducing the house edge in blackjack to less than one percent [116]. Phillips and colleagues conducted experiments where the blackjack program recommended the decision that was in accordance with Basic strategy to the user. The authors found that the presence of recommendations increased adherence to statistically optimal play, but also increased participants' willingness to make risky bets [97].

## Regulatory initiatives

Our charting suggested that the gambling research literature has empirically examined two main regulatory initiatives: restricting the supply of EGMs ($n = 2$) and restricting EGM features ($n = 2$). The supply reduction studies did not show favorable impact. Delfabbro [60] observed how restrictions in the number of EGMs allowed in venues in South Australia *increased* gambling revenues. A follow-up survey revealed that most gamblers in the region had noticed that the number of EGMs had fallen, but only a minority had reported gambling less as a result.

The feature restriction studies did show evidence of efficacy. Hansen and Rossow [69] found that adolescent gambling-related problems declined in Norway after the government removed banknote acceptors from slot machines. Participants reported having gambled fewer times in the past year after the ban on banknote acceptors. In a follow-up analysis [68], the authors reported that this decrease held across all levels of gambling, although the decrease in the proportion of participants gambling at least 80 times a year was about three times larger than the decrease in participants gambling at least 20 times a year.

## Replicability: Z-curve

The point estimates and 95% confidence intervals for the z-curve of the most focal hypothesis tests, as well as our 12 sensitivity tests (i.e., the highest *p*-values, the lowest *p*-values, and 10

**Table 3. Z-curve estimates and 95% confidence intervals.**

| Test | ODR | EDR | ERR | Max FDR | File-Drawer |
|------|-----|-----|-----|---------|-------------|
| Most Focal | .64 | .31 [.05, .75] | .65 [.47, .84] | .12 [.02, .94] | 2.28 [.34, 17.86] |
| Smallest *p* | .74 | .48 [.11, .95] | .79 [.63, .95] | .06 [.00, .42] | 1.10 [.06, 8.06] |
| Largest *p* | .50 | .10 [.05, .21] | .56 [.35, .74] | .46 [.20, 1.0] | 8.73 [3.84, 19.00] |
| Random 1 | .65 | .36 [.06, .78] | .63 [.43, .83] | .09 [.02, .81] | 1.77 [.28, 15.30] |
| Random 2 | .64 | .17 [.05, .60] | .65 [.47, .85] | .26 [.04, 1.00] | 4.89 [.67, 19.00] |
| Random 3 | .56 | .18 [.06, .73] | .73 [.55, .90] | .24 [.02, .81] | 4.64 [.37, 15.35] |
| Random 4 | .60 | .58 [.24, .88] | .75 [.54, .90] | .04 [.01, .17] | 0.72 [.14, 3.15] |
| Random 5 | .58 | .18 [.06, .82] | .74 [.55, .92] | .24 [.01, .77] | 4.62 [.22, 14.67] |
| Random 6 | .65 | .15 [.05, .57] | .60 [.42, .79] | .31 [.04, 1.0] | 5.85 [.75, 19.00] |
| Random 7 | .60 | .24 [.06, .68] | .67 [.49, .85] | .16 [.03, .91] | 3.11 [.48, 17.23] |
| Random 8 | .62 | .15 [.05, .59] | .69 [.50, .86] | .30 [.04, 1.00] | 5.72 [.69, 18.99] |
| Random 9 | .60 | .24 [.06, .68] | .67 [.49, .85] | .16 [.03, .91] | 3.11 [.48, 17.23] |
| Random 10 | .64 | .14 [.05, .32] | .67 [.50, .85] | .34 [.11, 1.0] | 6.36 [2.13, 19.00] |

Notes. All tests were calculated using a two-tailed alpha of .05. Test = Dataset on which z-curve was computed. ODR = Observed Discovery Rate. EDR = Expected Discovery Rate. ERR = Expected Replication Rate. Max FDR: Maximum False Discovery Rate; File-Drawer = Ratio of studies conducted to published significant findings.

iterations in which *p*-values from each study were randomly selected), are presented in Table 3. Because choosing the single most focal hypothesis test was difficult in many cases, we summarize the range of point estimates across the 13 iterations rather than focusing just on the single z-curve composed of *p*-values that we deemed most focal.

The Observed Discovery Rate ranged from .50 to .74 across our tests, indicating that most studies in the responsible product design literature report significant findings. The Expected Discovery Rate ranged from .10 to .58, suggesting that at least 42% of studies that have been conducted on responsible product design had null results. The Observed Discovery Rate was higher than the Expected Discovery Rate in all cases, by 204% on average. However, the confidence intervals of the Expected Discovery Rate were generally very large. In 8 of 13 iterations of z-curve (e.g., Most Focal and Smallest *p*), the upper confidence limit for the Expected Discovery Rate was higher than the point estimate of the Observed Discovery Rate, so this pattern is not statistically significant evidence of publication bias. In 4 cases (i.e., Largest *p* and Random 2), the point estimate for the Observed Discovery Rate was higher than the upper confidence limit of the Expected Discovery Rate. This is statistically significant evidence of publication bias. In one case (i.e., Random 4), the point estimate for the Observed Discovery Rate was well within the confidence interval of the Expected Discovery Rate and vice versa, providing some evidence against publication bias.

The Expected Replication Rate ranged from .60 to .79. The maximum false discovery rate ranged from .04 to .46. However, the confidence intervals for these estimates are so wide that they preclude our ability to make a general statement about the maximum proportion of significant findings that could be false positives. The file-drawer ratio ranged from 0.72 to 8.73. Similarly, the associated confidence intervals were too wide to allow us to draw conclusions about how many studies are conducted for each significant result that is published.

Re-running these 13 tests without studies that used a significance criterion other than a two-tailed alpha of .05 yielded a similar pattern of results, with the Observed Discovery Rate 149% larger than the Expected Discovery Rate on average. However, the Observed Discovery Rate was significantly higher than the Expected Discovery Rate in only one case (see S2 Table).

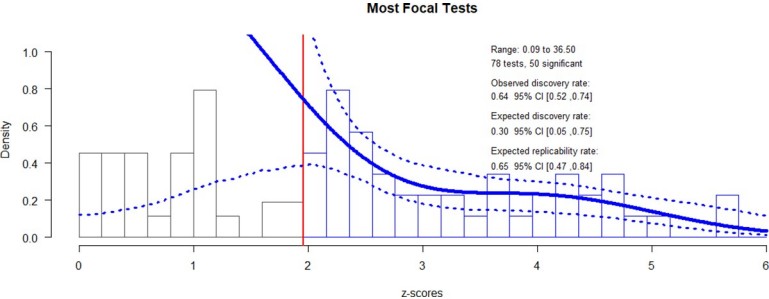

**Fig 3. Z-curve of most focal hypothesis tests.** The vertical red line refers to a z-score of 1.96, the critical value for statistical significance when using a two-tailed alpha of .05. The dark blue line is the density distribution for the inputted *p*-values (represented in the histogram as z-scores). The dotted lines represent the 95% confidence interval for the density distribution.

Fig 3 presents a histogram of *z*-scores from the most focal hypothesis tests, as well the estimates of replicability based on the z-curve of these values.

## Discussion

We conducted a scoping review of 86 studies evaluating game-based responsible gambling tools that were published between 2001 and 2020 to better understand the current state of the literature. Several general trends in study design were apparent in charting the included studies. First, studies were most likely to involve structural tools, followed by user-directed tools, and then game-specific regulations. (Of course, in practice some jurisdictions allow players to opt in or opt out of tools that we categorized as structural because the authors conceptualized and implemented them as involuntary. We only categorized tools as user-directed if participants could choose whether to use them or the authors noted that they would be implemented as user-directed.) Researchers were more likely to test structural features by experiments, about equally likely to use experiments or observational methods to test user-directed tools, and used only observational methods to study regulations.

The median sample size was 136. This is higher than in social and personality psychology (median = 104) [117] but lower than in clinical psychology (median = 179) [118] during comparable periods. Most studies did sample actual gamblers. Those that did not sampled from the community as often as they sampled from university participant pools, the latter of which is unrepresentative of individuals who are at risk of gambling harm [119]. Most studies did not include follow-up periods, but some of those that did measured gambling months later.

Most studies included self-reports, usually of gambling-related problems. However, most studies (61.6%) also included behavioral measures that were captured by gambling records. Given the importance of observing behavior to uncover people's preferences, responsible product design research is faring much better than social psychology, where only about 6%-12% of the empirical papers in what many regard as the field's premiere journal (*Journal of Personality and Social Psychology*) feature behavioral measures [120].

It was rare in the responsible product design literature or in narrative reviews thereof to find a discussion of cultural moderators of a given intervention's efficacy. Most interventions so far have been tested in a small number of countries, such as Canada and Australia. There were no studies which primarily sampled populations in Asia, Africa, or South America. These results are consistent with general trends in social science more broadly, and pose similar risks for overgeneralization [121]. Ideally, researchers would articulate the specific characteristics of their sample that theory would predict to constrain the generality of results [122].

## Preliminary conclusions about efficacy

Convergence in results across studies licenses some preliminary recommendations about which kinds of responsible product design are promising. Unfortunately, our review of the literature is consistent with earlier reviews: none of the game-based intervention tool types provide strong evidence for a particular strategy. Setting that reality aside, for structural feature tools, the best evidence supports pop-ups that encourage self-appraisal rather than pop-ups that attempt to rein in the influence of cognitive distortions. These pop-ups likely work in part because they create a brief break in play between trials. Imposing breaks long enough to effectively end a session, by contrast, increases craving and is irrelevant in settings where customers can easily switch to a different game. For covert interventions, modifications that undo features of games that promote excessive gambling likely have efficacy.

There are evidence-based reasons to doubt that user-directed tools are sufficient to prevent risky gambling. Many gamblers view pre-commitment tools as relevant only to people with gambling-related problems [123], and a common feature of experiencing such problems is a denial of excessive gambling [124]. Moreover, precommitment tools such as limit-setting and self-exclusion do not reliably reduce time or money spent gambling. On the other hand, ruling out riskier bets from the start [53] is a novel idea with some support. About a third of information aid studies indicated that they appear to have positive short-term effects, but long-term effects still require examination. Finally, with only four studies on regulatory initiatives, it is premature to draw conclusions about efficacy.

## Replicability and transparency

We tested whether the responsible product design literature contains publication bias using z-curve. Point estimates for the Expected Discovery Rate were on average much lower than Observed Discovery Rate point estimates. Thus, there is probably some publication bias based on statistical significance in the responsible product design literature. Bias based on statistical significance could manifest through not publishing null results or using questionable research practices to obtain significant results. However, the confidence intervals for the Expected Discovery Rate were very wide, leaving the magnitude of publication bias in this literature unclear. Furthermore, z-curve is a relatively new method, so we counsel caution in interpreting our results as the final word on the replicability of studies of product safety in gambling.

To the extent that direct replications inevitably differ in some respects from the original studies, the point estimates of the Expected Discovery Rate suggests that most replications of significant findings in the responsible product design literature would fail. Conditions appear more favorable, however, if exact replications could be conducted, as the Expected Replication Rate ranged from .61 to .79. These estimates are in line with large-scale replication efforts in experimental philosophy [125], experimental economics [126], and social science experiments published in *Science* and *Nature* [18], but lower than research on associations between the five-factor model of personality and consequential life outcomes [127].

How do our estimates of replicability compare with that of the Open Science Collaborative [19], which attempted to replicate 100 studies published in high-impact psychology journals in 2008 and has played a large role in generating concerns about low replicability in social science? Bartoš and Schimmack [32] constructed a z-curve on the original studies that were subjected to replication in the OSC, recovering an Observed Discovery Rate of .94 (95% CI [.87, .98]), an Expected Discovery Rate of .39 (95% CI [.07, .70]), and an Expected Replication Rate of .62 (95% CI [.46, .75]). The responsible product design literature has a greater tolerance for null results (or tests more true negatives), a slightly lower expected discovery rate (though the confidence intervals have considerable overlap), and a similar expected replicability rate.

These observations suggest that the literature on responsible product design in gambling is less insistent on the inclusion of significant findings than publications in eminent psychology journals were in 2008, but is not more insistent on replicable findings.

An exploratory analysis of z-curve within game-based intervention type did not lead to greater clarity. In fact, the z-curve would only run for studies of pop-up messages and information aids due to there being too few significant effects for other types of interventions. For pop-up messages, the results were very similar to the overall dataset. For information aids, the extant studies had a slightly lower Observed Discovery Rate, a lower Expected Discovery Rate, but very similar confidence interval, and a slightly lower Expected Replicability Rate. Hence, our appraisal of the replicability of responsible product design appears to generalize across intervention types, though this could be an artifact of pop-up messages making up the majority of interventions studied.

Effect size magnitude also can speak to replicability, as studies of large effects (all else equal) have more statistical power, but very large observed effects can be symptomatic of infidelities in the research process. In the responsible product design literature, researchers often report only standardized effect sizes. We recommend that researchers prioritize unstandardized effect sizes because they typically frame research questions in unstandardized terms [26], such as whether a certain tool will reduce money or time spent gambling. Even dependent variables that use rating scales, such as screeners for gambling-related problems, can be imbued with meaning because they often have validated thresholds based on harm severity [128]. Standardized effect sizes are appealing in large part because they put effects composed of different variables on the same metric. Nevertheless, standardized effect sizes do not directly illuminate the relative importance of different predictors because they are influenced by the variance of the predictors and the outcome [26, 129].

About one third of studies did not report any effect sizes. Journals could incentivize effect size reporting by insisting that researchers conduct power analyses. Few studies incorporated a power analysis, and about half of those that did reported a post-hoc power analysis, which is redundant with the *p*-value [129]. All included studies that did conduct an *a priori* power analysis used conventional benchmarks for what constitutes a medium or large effect [130]. Basing sample size on effect size conventions is dubious because it ignores the influence of measurement error [73]. Furthermore, small standardized effects may have large practical effects in the long-term or when scaled to a large population [27]. That said, we concur that power analysis should be based on the minimum effect size necessary to justify implementation. The smallest effect size of interest could be defined by the smallest change in the dependent variable that causes gamblers to report that they are experiencing less harm [131].

A lack of transparency undermines the benefit of adopting practices that increase replicability. Pre-registration limits obfuscation of which analyses were planned versus exploratory [24]. Although extant evidence does not support the contention that industry influences the methodology of responsible gambling research [132], pre-registration would be a worthwhile additional step to ensure independence between researchers and industry actors who fund them [133]. We found that only a handful of studies in the responsible product design literature were pre-registered, and all of them were published in 2019 or 2020. These findings are consistent with trends in psychological science, in which uptake of pre-registration between 2014 and 2017 was rare [134], perhaps because the studies on which such publications were based were conducted before knowledge of pre-registration was widespread.

Similarly, researchers must become more routine about disclosing their potential conflicts of interests and funding sources. About half of the included articles did not include a conflict of interest statement. Of course, journals have not always provided space to report conflicts of interest, or required a conflict of interest statement when they provide the space. Journals

must do their part in requiring authors to report all funding sources and potential conflicts of interest.

More generally, the absence of widespread transparent practices in the published gambling literature is not completely surprising, and we do not draw attention to this issue to single any-one out. Contemporary dialogue related to open science principles and practices became wide-spread in related academic sectors around 2012 (e.g., psychology [135, 136]); however, editorials addressing these topics to gambling researchers are more recent [137]. Supporting these editorials' calls for greater attention to open science with empirical evidence and trans-parent research practices should help advance meaningful change in gambling studies.

## Study limitations

The primary limitation of the present scoping review is its scope. Official reports and internal government studies on product safety in gambling were not included. We also did not include unpublished studies, some of which may have been high quality. This method is grounded in the conservative approach of keeping the review limited to research that has undergone the formal scrutiny of peer review. This decision was also consistent with the use of z-curve, which estimates the mean power of a published literature after selection on statistical significance. It is possible that including pre-prints, gray literature, and abandoned works would have allowed for a more direct assessment of the hypothesis that published results are more likely to report significant results than unpublished work. On the other hand, the difficulty of tracking down all relevant unpublished studies would risk underestimating the extent of publication bias in the peer-reviewed literature based on statistical significance.

A second limitation is that the keywords and inclusion criteria may have led us to exclude or overlook studies that would have changed our conclusions about efficacy or replicability. These potential omissions might have affected the conclusions we drew about replicability and publi-cation bias based on the results from z-curve. Third, a relatively small number of research groups have authored many of the included studies. The practices of those researchers might have a disproportionate influence on our inferences about how evaluations of game-based tools and regulations are designed, analyzed, and reported. Last, we could not include all eligible studies in z-curve because they did not include an inferential test of their most focal hypothesis.

## Conclusion

The responsible product design literature has several reassuring trends, such as widespread use of experimental methods and behavioral measures. But uncertainty about the literature's over-all methodological rigor, replicability and transparency precludes any strong recommenda-tions about which interventions stakeholders should promote and implement. Ignoring these important factors, currently the product safety literature provides the best evidence, albeit lim-ited evidence for pop-ups with self-appraisal messaging, breaks in between rounds of play, pre-commitment to less risky bets, undoing EGM features that promote excessive gambling, providing recommendations that minimize house edge, and removing banknote acceptors. Because the literature remains premature, we do not think that a meta-analysis on the respon-sible product design literature would settle this matter. Until there are a much larger number of high-powered, transparently reported studies, confident evidence-based product safety rec-ommendations remain elusive.

## Supporting information

**S1 Checklist. PRISMA 2009 checklist.**
(DOC)

**S1 Table.**
(TXT)

**S2 Table.**
(TXT)

## Acknowledgments

We thank Karen Amichia and Alessandra Grossman for their careful work in coding studies, and Matthew Tom for written feedback on previous drafts. We also express gratitude to Ulrich Schimmack, who provided extensive advice on conducting z-curve and interpreting its results.

## Author Contributions

**Conceptualization:** William H. B. McAuliffe, Timothy C. Edson, Eric R. Louderback, Debi A. LaPlante.

**Formal analysis:** William H. B. McAuliffe.

**Funding acquisition:** Debi A. LaPlante.

**Investigation:** William H. B. McAuliffe, Alexander LaRaja.

**Methodology:** William H. B. McAuliffe, Timothy C. Edson, Alexander LaRaja, Debi A. LaPlante.

**Project administration:** William H. B. McAuliffe, Eric R. Louderback, Alexander LaRaja, Debi A. LaPlante.

**Writing – original draft:** William H. B. McAuliffe, Eric R. Louderback.

**Writing – review & editing:** William H. B. McAuliffe, Timothy C. Edson, Eric R. Louderback, Alexander LaRaja, Debi A. LaPlante.

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
