## [Decision Letter · Decision Letter 0]

23 Dec 2020

PONE-D-20-27790

Responsible Product Design to Mitigate Excessive Gambling: A Scoping Review

PLOS ONE

Dear Dr. McAuliffe,

Thank you for submitting your manuscript to PLOS ONE. After careful consideration, we feel that it has merit but does not fully meet PLOS ONE’s publication criteria as it currently stands. Therefore, we invite you to submit a revised version of the manuscript that addresses the points raised during the review process.

Apologies for the delay in your review - we had multiple attempts to secure appropriate reviews.  As you will notice the reviews are somewhat mixed. The reviewers have raised concerns regarding appropriateness of the study design in relation to the aim of the study. Furthermore, the reviewers also believe that the limitations of the article has not been sufficiently discussed, as well as the presentation of results may be difficult for the readers to interpret. I am not sure that the authors will be able to address these concerns within the scope of major revision - however given the mixed reviews I would like to invite a resubmission. I would be grateful if you can take into consideration the aforementioned concerns when deciding whether to submit a revision. 

We look forward to receiving your revised manuscript.

Kind regards,

Simone Rodda

Academic Editor

PLOS ONE

Journal Requirements:

"This research will be supported primarily by a research contract between the Division on Addiction and GVC Holdings PLC (hereafter, GVC; https://gvc-plc.com/). GVC is a large international gambling and online gambling operator. GVC had no involvement with the development of our research questions or protocol. They will not see any associated materials (i.e., retrieved studies, charted data, and manuscripts in preparation) while the study is in progress or have any editorial rights to any resulting manuscripts. GVC communication about this work will require approval of the Division on Addiction" Please add this information as a COI within the online submission

Furthermore please also provide a table with the list of included studies within the manuscript text."

a. Please add this information to the Competing Interests section within the online submission form.

3. Please provide a table with the list of included studies within the manuscript text.

4.We note that this manuscript is a systematic review or meta-analysis; our author guidelines therefore require that you use PRISMA guidance to help improve reporting quality of this type of study. Please upload copies of the completed PRISMA checklist as Supporting Information with a file name “PRISMA checklist”.

Reviewers' comments:

Reviewer's Responses to Questions

**Comments to the Author**

1. Is the manuscript technically sound, and do the data support the conclusions?

Reviewer #1: Yes

Reviewer #2: Yes

Reviewer #3: Partly

Reviewer #4: Yes

Reviewer #5: Yes

2. Has the statistical analysis been performed appropriately and rigorously? 

Reviewer #1: Yes

Reviewer #2: Yes

Reviewer #3: Yes

Reviewer #4: Yes

Reviewer #5: Yes

3. Have the authors made all data underlying the findings in their manuscript fully available?

Reviewer #1: No

Reviewer #2: Yes

Reviewer #3: Yes

Reviewer #4: Yes

Reviewer #5: Yes

4. Is the manuscript presented in an intelligible fashion and written in standard English?

Reviewer #1: Yes

Reviewer #2: Yes

Reviewer #3: Yes

Reviewer #4: Yes

Reviewer #5: Yes

5. Review Comments to the Author

Reviewer #1: The manuscript reports a scoping review of consumer protection technologies directed towards the Gaming sector. Systematic reviews fall within PLOSone’s editorial guidelines, but I would have preferred a meta-analysis.

The focus of the study is power and statistical significance of quantitative research considering attempts to reduce gambling related harms. And concludes somewhat speciously there are some shortfalls in transparency. Speciously, since the same claims could be directed towards most sectors as these guidelines have only recently been introduced and implemented. An editorial could achieve a similar outcome.

The study is somewhat understated. Given that governments use the Gaming industry as means of raising revenues (i.e. a “stupidity tax”), there is an issue of accountability and there can be an obligation to minimise harm. Indeed, this is a sector where serious crimes such as embezzlement occur, and the perpetrators will oft claim “diminished responsibility”. For instance, there have been claims in court that gamblers were dissociated and lost track of time. As litigation is likely, responsible harm minimisation measures that are evidence based, peer reviewed, and generally accepted (cf. Daubert standard) may be required.

The manuscript is clearly quite systematic, but I struggled to identify “findings of interest”. But then I am not a fan of scoping reviews. A meta-analysis would better identify which interventions were efficacious. Even so, the tendency to clump a range of interventions together may mean that a powerful intervention can be overlooked if it is categorised with studies with poorer methodologies.

Specific points

Page 20 – reports both chi square and Fisher Exact test. Something requires clarification here. Fisher exact test is a one df test, whereas the chi square has 4 df.

Page 41 – there are definitely studies looking at irrational cognitions in Asian populations, but perhaps none that specifically address harm reduction methodologies that target these cognitions. Presumably because gambling would be illegal in many of these jurisdictions.

Page 34 – unstandardized effect sizes are not unstandardized Beta weights. For interpretability an unstandardized Beta requires information about intercepts which are rarely reported.

Reviewer #2: This is a review of responsible gambling interventions, with a unique focus on open science and replicability. Therefore, the review does have some attributes that warrant its publication. My comments are as follows.

1. The paper itself should include a summary table of all of the reviewed literature and some high-level features of each study. Although the online materials look good overall, the authors should not rely on the reader to use the materials to yield the review’s main conclusions. For example, starting on page 21 there is a categorization of the studies into intervention type, e.g. popup messages and breaks in play. However, this narrative review only covers some papers, and so many papers included in the review are not cited anywhere in the manuscript! This is unfortunate given the manuscript’s emphasis on openness and transparency. At present, I cannot reject the hypothesis that the papers actually cited in the review are restricted to the present authors’ “mates” --- based on my knowledge of this literature I certainly do not believe that the most replicable studies were selectively cited. Not citing these papers harms the average reader, and even harms the authors as their work will be harder to discover and will receive less attention.

2. The introduction to z-curve should give more background to these techniques and justify why z-curve was used instead of p-curve, which I believe is more commonly used.

3. The authors cite questionable research practices as a potential reason for low replicability. I think the authors should also explore other potential explanations. Biases in the published literature could also be due to rational responses to incentives. Null results papers are harder to publish, especially in high-impact journals. And not all authors have the financial resources to publish in PLOS One. Preprints are, in my view, the main way to counteract these biases. Now, uptake of preprints in gambling studies is likely low, but at least in principle allow authors a low-cost way to communicate their findings. If I am correct, then an expanded analysis should see a significant moderation of the z-curve results once preprints are included. This should be considered as a suggestion for future work in the Discussion.

Reviewer #3: Comments to authors

This paper is a scoping review of features that have been introduced to help minimise excessive gambling (so-called responsible gambling features). The authors have put a lot of work into this paper, and made their materials available via OSF, which in most cases is useful but in some cases I think that certain bits of information should be included in the actual paper instead (e.g., better descriptions of things coming from the Z curve analysis, because the descriptions of things like the file-drawer ratio aren’t great since most people won’t know this method, given how new it is). I hope that the comments below will be useful to the authors.

And before the authors read the comments below, I hope they understand that I appreciate the amount of work that has gone into this paper, especially in what seems to be a strong understanding of a novel technique.

The manuscript seems to be trying to do two things at once. First, it provides an overview of 86 studies that met inclusion criteria for this meta-analysis, in that those studies examined game-based features, tools or initiatives to help reduce excessive gambling. This part of the paper is generally done well, although discussion is lacking because it focuses more on the replicability components of the study. However, I have some concerns around the use of tests like correlations that are implicitly between-subjects, when the units of analysis (the studies) are clearly not independent of each other. For example, there are many papers from the same author(s), which of course would be similar to each other in terms of any relationship between variables, and would be likely to drive these results. This does not seem to be acknowledged.

I have some strong reservations about the Z curve analysis. Strong claims are made based on this novel technique. For example, I am uncomfortable with the claim that comparing the Observed Discovery Rate to the upper confidence limit of the Estimated Discovery Rate providing unambiguous evidence for publication bias. The way this is written on p13 (lines 222-223) suggests that a point estimate (the ODR) is compared to the confidence interval of the EDR, but the ODR has its own confidence interval, which does not seem to be taken into account. Indeed, comparing the confidence intervals throughout the iterations, the CI for the ODR and the EDR often overlap, which tells us nothing about differences (see https://towardsdatascience.com/why-overlapping-confidence-intervals-mean-nothing-about-statistical-significance-48360559900a or https://www.sciencedirect.com/science/article/pii/S0741521402000307 ). And this is my main problem with this section of the paper. This is a novel

technique that is not well known, and is not necessarily described well enough in the paper (although the authors made a valiant attempt!). The reader is instead required to read the supplementary materials to gain an understanding of the technique and the terms within. Having all of this in the supplementary material meant that this reader had to do a lot of work to understand this section of the manuscript, and the same would be true of other readers. I believe that this merits its own paper, with enough space spent in the methods to help a reader develop an understanding of this novel technique, rather than burying crucial information in online supplements.

My other concern about the Z curve analysis was initially that the citation was a preprint, which indicated that it hadn’t gone through peer review. It seems now that it has been published in Meta-Psychology, which lends the technique more credence, but I had to do the work to find that myself – the citation is still to a pre-print (it may be that this paper has been under review for awhile). I would have been very wary if the whole replicability part of the paper had been based on a preprint that had not at least gone through a peer review process. I’m still wary about early adoption of novel techniques that are not well understood, when very strong claims are made, or would have been made if the results were more conclusive, without appropriate acknowledgement of this as a limitation. In fact, the limitations study is lacking several important limitations, that I have outlined in this review.

I also think there’s a large discrepancy between the results section (which are mostly the first part of the study, referring to efficacy, and a little bit on replicability) and the discussion (where most of the discussion is about replicability, and there’s little about efficacy). And this leads to my recommendation below.

There is a subtle but somewhat pervasive issue with the way certain things are reported. For example, studies were classified based on certain things, e.g., whether there was a conflict of interest statement. There is an implication that this is an omission on the part of the authors, and this becomes very clear at the end of the discussion. I think it is important to recognise that many journals historically haven’t required this, and some don’t even have space at all for it, which is why it’s not there. Other journals require these statements now, which is why they may be there now. I think things like this should be acknowledged to provide a more balanced presentation of these results, especially since the authors appear to blame the authors for this later in the paper. To highlight this point further, preregistration is indicated as being quite new (e.g., the finding that all of the preregistered studies were in 2019 or 2020), but this doesn’t seem to have been looked at

for conflict of interest statements. In my experience, journals are moving more towards these types of requirements. So I’m not convinced that authors should be instructed to become more routine about disclosing their potential conflicts of interest and funding sources, without acknowledging that the journal may not have had space for that at the time of publication, or that they may not have been required before but are now, and that their exclusion is not necessarily the author’s fault.. Further, certain classifications like “abstract deemed vague” are a little… condescending, really. I’m aware that this may feel like tone policing, and I apologise if it comes across like that, but I hope the authors can see where I’m coming from.

Overall, I think this paper would be far improved by splitting it into two papers, rather than trying to do too many things in one paper. (And of course I am aware of salami slicing concerns, but I think these are two very distinct messages.) I think the first part of the results, that describes the meta-analysis and efficacy, is strong enough to stand alone, but needs more space in the discussion in particular. I think the second part is not described in enough detail in the methods to be interpretable for most readers, and therefore requires far more explanation than is in the actual paper itself. As the paper currently stands, it feels like the authors set out to look at efficacy, but actually they really want to talk about open science, and this is why I think these things are two quite separate papers, with each deserving its own space to be fully explored and discussed.

But to be clear, it seems like the authors have done a good job with this work, and I commend them making all of their material available online, including data and analysis scripts. It’s just that the way that it’s written doesn’t work very well for the reader, in my opinion, and that the paper overall needs a more singular focus. I do hope to see these data published in the near future, and I hope these comments help the authors to make this happen.

Specific comments:

Line 279 - Some studies are described as experimental, but then some of those don’t involve random assignment of participants to condition. For most readers, experiment = random assignment, so what is it about the 5 studies that make them experimental without involving random assignment?

Line 341-342 - Did any studies report BOTH unstandardised and standardised effect sizes? This kind of issue potentially occurs with a lot of the classifications, where there may be overlap, so the authors might consider this throughout.

Line 371 - This may be a personal preference, but terms like “marginally significant” are, in my opinion, pretty ordinary. Similarly later on where the authors discuss results being suggestive but not conclusive.

Line 395 - 2020 papers apparently go to May, but the database search was in February. I understand the authors included some other papers that were in reference lists, but this needs to be actually explained at this point to explain this apparent discrepancy.

Reviewer #4: Overall, the study is well executed in terms of scope, method and language. Several reviews have been carried out in recent years. However, this current scoping review address an important topic that the previous reviews point towards. Investigating how studies have been carried and the possibility to replicate is important! As the authors have noted, there is a lack of studies and many of lack scientific. I think the current study adds a lot to the discussion about responsible gambling on a general level.

However, some minor changes need to be made. There some minor mistakes in the manuscript, e.g. double parenthesis. The manuscript should be checked for those types of mistake. However, this is a minor problem.

Introduction

The first paragraph of the introduction can be deleted. The paragraph is not necessary. A focus on addiction is perhaps better. A part from that the introduction is well-written and presents the subject and the need for the in a good.

came to similar conclusions that was reached in the scoping review. However, the examination of the studies was done in a less rigorous way than in the scoping review. The meta-analysis can be used both in the intro to better show the importance of the aim of the scoping review and in the discussion when it comes to transparency and the effectiveness of the interventions.

Methods

The way the study was carried out and described is good. The use of PRISMA makes is it easy to understand the selection of the studies.

The pre-registration and reliability analysis of the raters is very good and shows the scientific rigor of the researchers responsible for the study.

One thing that could be added is how this type of analysis has been carried out in the field of addiction would be interesting and also what types of benchmarks thy have used.

Also, the description of the analytic strategy covers most of the analysis done. The authors have not included a reference or a description of the tetrachoric correlations and why it was in the Methods section. Information about Fisher’s exact test is also missing.

Results

The results are presented in a clear way. It is long and perhaps it is possible to make it a bit shorter.

Discussion

Overall, the discussion is well-written and covers most of the presented results. The discussion about the origin of most of the studies included is important and rarely discussed in the field of gambling research.

The discussion about should be elaborated using Ladouceur, Shaffer, Blaszczynski, & Shaffer (2019). Responsible gambling research and industry funding biases. Journal of gambling studies, 35(2), 725-730. It can also be discussed in relation to transparency a bit more. What are the potential consequences of this lack of transparency?

Reviewer #5: An interesting paper that provides insight into the evaluation of responsible gambling features to date and sets out some sound principles for best practice going forward. I am not sure that the term "gambler" is appropriate for describing all peoples that gamble, as it has negative connotations and may not reflect the self perception on many who gamble from time to time. The term "player" is less value laden. Also, I believe the EGM ban in Norway was in July 2007 not 2006.

6. PLOS authors have the option to publish the peer review history of their article (what does this mean?). If published, this will include your full peer review and any attached files.

Reviewer #1: No

Reviewer #2: No

Reviewer #3: No

Reviewer #4: No

Reviewer #5: No

---

## [Author Response · Author response to Decision Letter 0]

28 Jan 2021

Response Letter for PONE-D-20-27790-R1

Dear Dr. Rodda,

Thank you for the opportunity to submit a revised version of “Responsible Product Design to Mitigate Excessive Gambling: A Scoping Review” to PLOS ONE. We greatly appreciate the insightful comments on the manuscript from you and the referees. We describe below how we responded to each reviewer comment. We thank the reviewers for helping us strengthen the manuscript and believe that the revised version is much improved. 

As you requested, we added a table to the manuscript with all study citations, and attached a completed PRISMA checklist. To conform to PRISMA guidelines, we changed the formatting of the abstract to a structured summary. Noelle Gibbs also asked us to include the following statement in the cover letter:

This research was supported primarily by a research contract between the Division on Addiction and GVC Holdings PLC (now called Entain; https://entaingroup.com/). Entain is a large international gambling and online gambling operator. Entain had no involvement in the development of our research questions or protocol. They will not see any associated materials (i.e., retrieved studies, charted data, and manuscripts in preparation) while the study is in progress or have any editorial rights to any resulting manuscripts. Entain communication about this work will require approval of the Division on Addiction.

We have uploaded revised versions of the manuscript to the PLOS ONE manuscript submission portal, including a version with highlighted changes and a version with the changes but without the highlighting. 

Reviewer #1: The focus of the study is power and statistical significance of quantitative research considering attempts to reduce gambling related harms. And concludes somewhat speciously there are some shortfalls in transparency. Speciously, since the same claims could be directed towards most sectors as these guidelines have only recently been introduced and implemented. An editorial could achieve a similar 

outcome.

Response: We agree that some aspects of transparency have gained popular attention only somewhat recently (e.g., open data and analytic code, pre-registration, etc.), but also note that some of the metrics on which we are evaluating studies have been discussed for a long time. In 1962, Cohen wrote “It is further recommended that research plans be routinely subjected to power analysis” (p. 153). Likewise, the 5th edition of the APA manual, published in 2001, regarded the failure to report effect sizes as a “defect in the design and reporting of research” (p. 5). Finally, The Transparency Project (www.thetransparencyproject.org) has advocated for Open Science-informed data practices in gambling studies research for more than a decade. That said, we have added to the discussion an explicit acknowledgement that some standards are novel: 

“The absence of widespread transparent practices in the published gambling literature is not completely surprising, and we do not draw attention to this issue to single anyone out. Contemporary dialogue related to open science principles and practices became widespread in related academic sectors around 2012 (e.g., in psychology, 77,78); however, editorials addressing these topics to gambling researchers are more recent (79). Supporting these editorials' calls for greater attention to open science with empirical evidence and transparent research practices should help advance meaningful change in gambling studies.”

We also added an acknowledgement that recent adoption of pre-registration is understandable, given that is has only recently become mainstream:

“We found that only a handful of studies in the responsible product design literature were pre-registered, and all of them were published in 2019 or 2020. These findings are consistent with trends in psychological science, in which uptake of pre-registration between 2014 and 2017 was rare (78), perhaps because the studies on which such publications were based were conducted before knowledge of pre-registration was widespread.”

American Psychological Association. (2001). Publication manual of the American Psychological Association (5th ed.). Washington, DC: Author

Cohen, J. (1962). The statistical power of abnormal–social psychological research: A review. Journal of Abnormal and Social Psychology, 65, 145–153. doi:10.1037/h0045186

Reviewer 1: The manuscript is clearly quite systematic, but I struggled to identify “findings of interest”. But then I am not a fan of scoping reviews. A meta-analysis would better identify which interventions were efficacious. Even so, the tendency to clump a range of interventions together may mean that a powerful intervention can be overlooked if it is categorised with studies with poorer methodologies.

Response: We did not opt for a meta-analysis in the present study for two reasons. First, very few intervention types had enough empirical studies, limiting the feasibility and usefulness of a meta-analysis. Second, methodological limitations in many of the included studies would render the interpretation of average effects relatively meaningless, as Reviewer 1 alludes. More fundamentally, though, this scoping review is focused on examining the scientific practices of researchers who conduct studies on product safety tools for gambling. Even if the average effects of all interventions could be meta-analytically estimated without bias, it is still relevant to know whether researchers pre-register, consistently report effect sizes, conduct a priori power analyses, publish studies that can be trusted to replicate, and so on. 

Reviewer 1: Page 20 – reports both chi square and Fisher Exact test. Something requires clarification here. Fisher exact test is a one df test, whereas the chi square has 4 df.

Response: Thank you for raising this methodological point. Fisher’s exact test’s p-value does not depend on degrees of freedom. The chi-square test’s p-value is based on (c-1)*(r-1) degrees of freedom. We have clarified in the manuscript that the use of Fisher’s exact test was an exploratory follow-up to account for low expected cell counts (see Larntz, 1978).

Larntz, Kinley (1978). "Small-sample comparisons of exact levels for chi-squared goodness-of-fit statistics". Journal of the American Statistical Association. 73 (362): 253–263.

Reviewer 1: Page 41 – there are definitely studies looking at irrational cognitions in Asian populations, but perhaps none that specifically address harm reduction methodologies that target these cognitions. Presumably because gambling would be illegal in many of these jurisdictions.

Response: We have added the following text to the “Unplanned Coding of Sampled Populations” section of the results:

“Though studies examining causes of excessive gambling in Asian, African, or South American countries exist (e.g., 38,39), none of the included studies sampled these populations.”

Reviewer 1: Page 34 – unstandardized effect sizes are not unstandardized Beta weights. For interpretability an unstandardized Beta requires information about intercepts which are rarely reported.

Response: We agree that researchers should report both intercepts and slopes in regression tables. We do not believe that we suggested that unstandardized effect sizes are unstandardized Beta weights.

Reviewer #2: The paper itself should include a summary table of all of the reviewed literature and some high-level features of each study. Although the online materials look good overall, the authors should not rely on the reader to use the materials to yield the review’s main conclusions. For example, starting on page 21 there is a categorization of the studies into intervention type, e.g. popup messages and breaks in play. However, this narrative review only covers some papers, and so many papers included in the review are not cited anywhere in the manuscript! This is unfortunate given the manuscript’s emphasis on openness and transparency. At present, I cannot reject the hypothesis that the papers actually cited in the review are restricted to the present authors’ “mates” --- based on my knowledge of this literature I certainly do not believe that the most replicable studies were selectively cited. Not citing these papers harms the average reader, and even harms the authors as their work will be harder to discover and will receive less attention.

Response: We appreciate this suggestion and have added a table (Table 2) to the main text with all included studies so that readers do not need to consult the Supplemental Materials. Unfortunately, it would not be possible for us to highlight the “most replicable” studies in our narrative review because the replicability of any one study depends on factors we cannot directly observe, such as the true effect size. Until direct replications are conducted, we can only estimate the overall replicability of a body of literature using tools like z-curve. We note, however, that we did not go out of our way to highlight studies with methodological limitations or other shortcomings in design. For instance, we described the protocol and findings of Ivanova et al. (2019), which we described as the “best-designed study” in the literature, to justify our narrative explanation associated with the subset of precommitment studies showing null or unfavorable results. 

Reviewer #2: The introduction to z-curve should give more background to these techniques and justify why z-curve was used instead of p-curve, which I believe is more commonly used.

Response: We agree that the z-curve analytic approach might not be widely known among readers, so we appreciate this suggestion to add more details regarding this methodology. We moved text explaining z-curve from our online supplement to the z-curve sub-section of the methods section. In the “Replicability of Responsible Product Design Research” section of the introduction, we now explicitly mention our rationale for preferring z-curve: “Brunner and Schimmack (22) find that z-curve outperforms p-curve, p-uniform, and maximum likelihood estimation in estimating mean power of a set of studies selected for significance when there is heterogeneity in effect sizes (pgs. 12-13). We expect heterogeneity in effect sizes because different researchers are studying the effects of different types of interventions.”

Reviewer 2: The authors cite questionable research practices as a potential reason for low replicability. I think the authors should also explore other potential explanations. Biases in the published literature could also be due to rational responses to incentives. Null results papers are harder to publish, especially in high-impact journals. And not all authors have the financial resources to publish in PLOS One. Preprints are, in my view, the main way to counteract these biases. Now, uptake of preprints in gambling studies is likely low, but at least in principle allow authors a low-cost way to communicate their findings. If I am correct, then an expanded analysis should see a significant moderation of the z-curve results once preprints are included. This should be considered as a suggestion for future work in the Discussion.

Response: We heartily agree that both questionable research practices and publication bias contribute to low replicability. It is possible that consideration of pre-prints or the unpublished grey literature might improve estimates of the replicability of gambling research; however, the central purpose of z-curve is to test for selection processes in the published literature. The comparison of the Observed Discovery Rate to the Expected Discovery Rate is a test for whether null results have been “harder to publish.” We have added the following statement to the limitations section of the discussion: 

“We also did not include unpublished studies, some of which might have been of high quality. This method is grounded in the conservative approach of keeping the review limited to research that has undergone the formal scrutiny of peer review. This decision was also consistent with the use of z-curve, which estimates the mean power of a published literature after selection on statistical significance. It is possible that including pre-prints, gray literature, and abandoned works would have allowed for a more direct assessment of the hypothesis that published results are more likely to report significant results than unpublished work. On the other hand, the difficulty of tracking down all relevant unpublished studies would risk underestimating the extent of publication bias in the published literature based on statistical significance.”

Reviewer 3: The manuscript seems to be trying to do two things at once. First, it provides an overview of 86 studies that met inclusion criteria for this meta-analysis, in that those studies examined game-based features, tools or initiatives to help reduce excessive gambling. This part of the paper is generally done well, although discussion is lacking because it focuses more on the replicability components of the study. However, I have some concerns around the use of tests like correlations that are implicitly between-subjects, when the units of analysis (the studies) are clearly not independent of each other. For example, there are many papers from the same author(s), which of course would be similar to each other in terms of any relationship between variables, and would be likely to drive these results. This does not seem to be acknowledged.

Response: Thank you for raising these points. We followed an established scoping review methodology in our article, which is designed to review, synthesize, and identify gaps in the literature in a particular area of study (for more details on the scoping review methodology, see Arksey & O'Malley, 2005; Munn et al., 2018; Peters et al., 2015). With regards to overlap in the authorship of included studies, we added the following to the limitations section of the discussion: 

“...Third, a relatively small number of research groups have authored many of the included studies. The practices of those researchers might have a disproportionate influence on our inferences about how evaluations of game-based tools and regulations are designed, analyzed, and reported.”

Arksey, H., & O'Malley, L. (2005). Scoping studies: towards a methodological framework. International journal of social research methodology, 8(1), 19-32.

Munn, Z., Peters, M. D., Stern, C., Tufanaru, C., McArthur, A., & Aromataris, E. (2018). Systematic review or scoping review? Guidance for authors when choosing between a systematic or scoping review approach. BMC medical research methodology, 18(1), 143.

Peters, M. D., Godfrey, C. M., Khalil, H., McInerney, P., Parker, D., & Soares, C. B. (2015). Guidance for conducting systematic scoping reviews. International journal of evidence-based healthcare, 13(3), 141-146.

Reviewer 3: I have some strong reservations about the Z curve analysis. Strong claims are made based on this novel technique. For example, I am uncomfortable with the claim that comparing the Observed Discovery Rate to the upper confidence limit of the Estimated Discovery Rate providing unambiguous evidence for publication bias. The way this is written on p13 (lines 222-223) suggests that a point estimate (the ODR) is compared to the confidence interval of the EDR, but the ODR has its own confidence interval, which does not seem to be taken into account. Indeed, comparing the confidence intervals throughout the iterations, the CI for the ODR and the EDR often overlap, which tells us nothing about differences (see https://towardsdatascience.com/why-overlapping-confidence-intervals-mean-nothing-about-statistical-significance-48360559900a or https://www.sciencedirect.com/science/article/pii/S0741521402000307 ). And this is my main problem with this section of the paper.

Response: Thank you for raising these points regarding the z-curve methodology. We agree that two estimates can be statistically significantly different from one another when their confidence intervals overlap. We did not infer statistical significance based on whether there was overlap between the confidence intervals of the Expected Discovery Rate and Observed Discovery Rate. We inferred statistical significance when the point estimate of the Observed Discovery Rate was higher than the upper confidence limit of the Expected Discovery Rate.

However, we agree that we did not explicitly articulate the basis on which we made determinations about the presence of publication bias. We asked the developer of z-curve, Dr. Ulrich Schimmack, via e-mail correspondence about the clearest way to compare the Observed Discovery Rate and Expected Discovery Rate. In accordance with his advice, we have lumped together all 4 cases in which the point estimate of the Observed Discovery Rate is higher than the upper confidence limit of the Expected Discovery Rate, and have declared these as the only 4 cases in which the two are statistically significantly different. We also note that the Observed Discovery Rate is higher than the Expected Discovery Rate in all cases, and report the average percentage difference between these two estimates. As in our original manuscript, we emphasize that the Expected Discovery Rate was estimated only very imprecisely. Finally, we have eliminated the confidence intervals around the Observed Discovery Rate to underline that this parameter is not estimated but known from the analysis of the included studies. The confidence interval is printed by the zcurve package on the assumption that the test statistics are a sample from a larger population of studies that would have met inclusion criteria. Dr. Schimmack reminded us that our included studies represent the entire population of interest. 

The results section for z-curve now reads:

“The Observed Discovery Rate ranged from .50 to .74 across our tests, indicating that most studies in the responsible product design literature report significant findings. The Expected Discovery Rate ranged from .10 to .58, suggesting that at least 42% of studies that have been conducted on responsible product design had null results. The Observed Discovery Rate was higher than the Expected Discovery Rate in all cases, by 204% on average. However, the confidence intervals of the Expected Discovery Rate were generally very large. In 8 of 13 iterations of z-curve (e.g., Most Focal and Smallest p), the upper confidence limit for the Expected Discovery Rate was higher than the point estimate of the Observed Discovery Rate, so this pattern is not statistically significant evidence of publication bias. In 4 cases (i.e., Largest p and Random 2), the point estimate for the Observed Discovery Rate was higher than the upper confidence limit of the Expected Discovery Rate. This is statistically significant evidence of publication bias. In one case (i.e., Random 4), the point estimate for the Observed Discovery Rate was well within the confidence interval of the Expected Discovery Rate and vice versa, providing some evidence against publication bias. 

The Expected Replication Rate ranged from .60 to .79. The maximum false discovery rate ranged from .04 to .46. However, the confidence intervals for these estimates are so wide that they preclude our ability to make a general statement about the maximum proportion of significant findings that could be false positives. The file-drawer ratio ranged from 0.72 to 8.73. Similarly, the associated confidence intervals were too wide to allow us to draw conclusions about how many studies are conducted for each significant result that is published. 

Re-running these 13 tests without studies that used a significance criterion other than a two-tailed alpha of .05 yielded a similar pattern of results, with the Observed Discovery rate 149% larger than the Expected Discovery Rate on average. However, the Observed Discovery Rate was significantly higher than the Expected Discovery Rate in only case (see Table S2 at https://osf.io/8merf/)."

Reviewer #3: “…in some cases I think that certain bits of information should be included in the actual paper instead (e.g., better descriptions of things coming from the Z curve analysis, because the descriptions of things like the file-drawer ratio aren’t great since most people won’t know this method, given how new it is).”

“This [z-curve] is a novel technique that is not well known, and is not necessarily described well enough in the paper (although the authors made a valiant attempt!). The reader is instead required to read the supplementary materials to gain an understanding of the technique and the terms within. Having all of this in the supplementary material meant that this reader had to do a lot of work to understand this section of the manuscript, and the same would be true of other readers. I believe that this merits its own paper, with enough space spent in the methods to help a reader develop an understanding of this novel technique, rather than burying crucial information in online supplements.”

Response: We appreciate this helpful and insightful feedback. We have moved our introduction to z-curve from the supplement to the main text, which now reads:

“We used the zcurve package in R to conduct z-curve analyses (33). Z-curve is based on the idea that a distribution of z-scores can be derived from the average power of an entire set of studies. That distribution is truncated at the critical z-value (typically 1.96) after selection for statistical significance. Z-curve takes as input the set of significant findings (to mimic the editorial process of publishing only positive findings) and uses this truncated distribution to estimate the most likely shape of the non-truncated distribution of the population represented by the significant studies. To account for heterogeneity in effect sizes and power, z-curve estimates the distribution of all conducted studies using a finite mixture model of seven distributions, centered on z-scores of 0,1, 2, 3, 4, 5, and 6, respectively. An expectation maximization algorithm is used to assign studies probabilities of belonging to each distribution (34). 

The resulting estimate of the non-truncated z-score distribution enables the computation of several statistics. First, the area under the curve to the right of the significance criterion is the Estimated Discovery Rate, or the estimated proportion of all studies that have been conducted that had significant results. The Observed Discovery Rate represents the proportion of coded tests that had significant results in the hypothesized direction. Because our dataset represents the entire population of interest, we omit confidence intervals from our reports of the Observed Discovery Rate. Evidence for publication bias exists if the Observed Discovery Rate is higher than the upper confidence limit of the Estimated Discovery Rate. 

The Estimated Discovery Rate can be used to estimate how many non-significant results there might be for each significant result. This “file-drawer ratio” is equal to the estimated proportion of non-significant results (1- Estimated Discovery Rate) divided by the Estimated Discovery Rate. The file-drawer ratio can in turn be used to compute the False Discovery Risk, or the maximum proportion of significant studies that could represent false positives. The False Discovery Risk equals the product of the file-drawer ratio and the ratio of alpha (viz., .05) to 1-alpha. 

Finally, the Expected Replication Rate is the mean power of the non-truncated distribution, and represents the estimated proportion of significant studies that would yield another significant effect if subjected to a direct replication. However, commentators frequently point to differences between original studies and replication studies that could explain why the former yield larger effect sizes than the latter (34, 35). Consequently, the Expected Discovery Rate should more accurately predict the outcome of actual replication efforts than the Expected Replication Rate.”

Reviewer 3: My other concern about the Z curve analysis was initially that the citation was a preprint, which indicated that it hadn’t gone through peer review. It seems now that it has been published in Meta-Psychology, which lends the technique more credence, but I had to do the work to find that myself – the citation is still to a pre-print (it may be that this paper has been under review for awhile). I would have been very wary if the whole replicability part of the paper had been based on a preprint that had not at least gone through a peer review process. I’m still wary about early adoption of novel techniques that are not well understood, when very strong claims are made, or would have been made if the results were more conclusive, without appropriate acknowledgement of this as a limitation.

Response: The z-curve paper was in press at the time we worked on this project. The typeset version of it was just not ready when we submitted this manuscript. However, the level of scrutiny that z-curve has received goes well beyond peer-review. The method has been extensively discussed among academics since 2015 (e.g., see page 14 of Brunner and Schimmack, where they respond to a published comment from 2018), and has been used in published papers both before and after Brunner and Schimmack (2020) finally appeared in print:

Motyl, M., Demos, A. P., Carsel, T. S., Hanson, B. E., Melton, Z. J., Mueller, A. B., ... & Yantis, C. (2017). The state of social and personality science: Rotten to the core, not so bad, getting better, or getting worse?. Journal of personality and social psychology, 113(1), 34.

Schimmack, U. (2020). A meta-psychological perspective on the decade of replication failures in social psychology. Canadian Psychology/Psychologie canadienne, 61(4), 364–376. https://doi.org/10.1037/cap0000246

All of that said, we have added the following to the “Replicability and Transparency” section of the discussion:

“Furthermore, z-curve is a relatively new method, so we counsel caution in interpreting our results as the final word on the replicability of studies of product safety in gambling.”

Reviewer 3: There is a subtle but somewhat pervasive issue with the way certain things are reported. For example, studies were classified based on certain things, e.g., whether there was a conflict of interest statement. There is an implication that this is an omission on the part of the authors, and this becomes very clear at the end of the discussion. I think it is important to recognise that many journals historically haven’t required this, and some don’t even have space at all for it, which is why it’s not there. Other journals require these statements now, which is why they may be there now. I think things like this should be acknowledged to provide a more balanced presentation of these results, especially since the authors appear to blame the authors for this later in the paper. To highlight this point further, preregistration is indicated as being quite new (e.g., the finding that all of the preregistered studies were in 2019 or 2020), but this doesn’t seem to have been looked at

for conflict of interest statements. In my experience, journals are moving more towards these types of requirements. So I’m not convinced that authors should be instructed to become more routine about disclosing their potential conflicts of interest and funding sources, without acknowledging that the journal may not have had space for that at the time of publication, or that they may not have been required before but are now, and that their exclusion is not necessarily the author’s fault..

Response: This is a fair point, and in the “Replicability and Transparency” section of the discussion we added the following: “Of course, journals have not always provided space to report conflicts of interest, or required a conflict of interest statement when they provide the space. Journals must do their part in requiring authors to report all funding sources and potential conflicts of interest.”.

Reviewer 3: Further, certain classifications like “abstract deemed vague” are a little… condescending, really. I’m aware that this may feel like tone policing, and I apologise if it comes across like that, but I hope the authors can see where I’m coming from.

Response: We did not intend for the term “vague” to be interpreted as an admonishment, and apologize if using this term came off as condescending. We acknowledge that authors have no idea what systematic reviews future authors will conduct when they publish scientific research, and cannot tailor their abstracts in such a way that they can be conveniently examined against all possible inclusion criteria. To avoid any confusion, we have replaced instances of “vague study titles and abstracts” with “When it was not clear from the title and abstract alone whether a paper met inclusion criteria, we retained the full text for inspection.”

Reviewer 3: I also think there’s a large discrepancy between the results section (which are mostly the first part of the study, referring to efficacy, and a little bit on replicability) and the discussion (where most of the discussion is about replicability, and there’s little about efficacy)...Overall, I think this paper would be far improved by splitting it into two papers, rather than trying to do too many things in one paper. (And of course I am aware of salami slicing concerns, but I think these are two very distinct messages.) I think the first part of the results, that describes the meta-analysis and efficacy, is strong enough to stand alone, but needs more space in the discussion in particular. I think the second part is not described in enough detail in the methods to be interpretable for most readers, and therefore requires far more explanation than is in the actual paper itself. As the paper currently stands, it feels like the authors set out to look at efficacy, but actually they really want to talk about open science, and this is why I think these things are two quite separate papers, with each deserving its own space to be fully explored and discussed.

Response: We appreciate the assessment that this study covers a lot of ground. We were more limited in what we could say about efficacy than we would have liked because of the low quality of studies and small number of studies on any one type of tool. But what we were primarily interested in is not efficacy per se but the overall methodological quality and scientific rigor of the manuscripts. This emphasis is now underlined in the introduction:

“Researchers have called for a greater emphasis on implementing safety features and interventions for gambling products (4). However, it would be premature to make implementation recommendations without first determining whether existing evidence is based on sound research practices.”

We also suggest that evaluations of efficacy cannot be separated from considerations of replicability because if we have limited confidence that a study finding will replicate, then there is little reason to draw conclusions about efficacy from the study’s findings. Meta-analysis can only make up for the low power of individual studies if there is little or no publication bias or questionable research practices (e.g., post-hoc or non-existent power analysis, p-hacking, very selective outcome reporting, see Bishop, 2019 in Nature for more details). Similarly, open science and efficacy are not independent because non-transparently conducted research might result in overly confident conclusions. By analogy, a recent review of longitudinal studies on whether vaping is a “gateway” to cigarette smoking (https://onlinelibrary.wiley.com/doi/10.1111/add.15246) noted that most study authors failed to report predictors of attrition and account for them in reporting study results. The authors correctly point out that this omission makes drawing substantive conclusions difficult because, for all the reader knows, there is overlap in the characteristics that cause study attrition (i.e., drop-out) and the initiation of cigarette smoking, resulting in biased causal estimates.

Bishop, D. (2019). Rein in the four horsemen of irreproducibility. Nature, 568(7753).

Reviewer 3: Line 279 - Some studies are described as experimental, but then some of those don’t involve random assignment of participants to condition. For most readers, experiment = random assignment, so what is it about the 5 studies that make them experimental without involving random assignment?

Response: This is an astute observation. We have modified our summary of study design to read: “We observed that 69.8% (n = 60) of all studies were experimental, 91.7% (n = 55) of which randomly assigned participants to condition (i.e., true experiments). The other 5 studies were quasi-experimental in that they contained multiple conditions but did not randomly assign participants to conditions. "

Reviewer 3: Line 341-342 - Did any studies report BOTH unstandardised and standardised effect sizes? This kind of issue potentially occurs with a lot of the classifications, where there may be overlap, so the authors might consider this throughout.

Response: We agree this is important information and have now clarified how many studies reported both types of effect sizes: “37.0% (n = 20) reported at least one unstandardized effect size, 77.8% (n = 42) reported at least one standardized effect size, and 9.3% (n = 8) reported at least one unstandardized effect and at least one standardized effect size.”

Reviewer 3: Line 371 - This may be a personal preference, but terms like “marginally significant” are, in my opinion, pretty ordinary. Similarly later on where the authors discuss results being suggestive but not conclusive.

Response: Thank you for commenting on this terminology. We have changed both “marginally significant” and “suggestive but not conclusive” to “nonsignificant.”

Reviewer 3: Line 395 - 2020 papers apparently go to May, but the database search was in February. I understand the authors included some other papers that were in reference lists, but this needs to be actually explained at this point to explain this apparent discrepancy.

Response: On page 8 we wrote “Our analytic sample consisted of all of the eligible studies from the database search (N = 43), as well as (N = 23) studies that the first author found after examining the reference lists of previous reviews, the studies that met inclusion criteria, and studies that cited the included studies according to Google Scholar.” In the revised manuscript, we noted that the discovery of the 23 additional studies occurred between February 2020 and May 2020. 

Reviewer #4: There some minor mistakes in the manuscript, e.g. double parenthesis. The manuscript should be checked for those types of mistake. However, this is a minor problem.

Response: Thanks for pointing this out. We closely read over the revised manuscript several times and fixed these typos.

Reviewer #4: The first paragraph of the introduction can be deleted. The paragraph is not necessary. A focus on addiction is perhaps better. 

Response: We reviewed the paragraph identified in this comment and have elected to keep this paragraph in the revised manuscript. We find that the opening analogy is helpful to readers who might not be intimately familiar with responsible product design for gambling. 

Reviewer 4: came to similar conclusions that was reached in the scoping review. However, the examination of the studies was done in a less rigorous way than in the scoping review. The meta-analysis can be used both in the intro to better show the importance of the aim of the scoping review and in the discussion when it comes to transparency and the effectiveness of the interventions.

Response: Reviewer #4’s comment here is unclear (and potentially incomplete) therefore we did not make any changes. We also point this reviewer to our response above to Reviewer 3 regarding the approach and goals of the scoping review methodology, as well as the three articles cited in that response.

Reviewer 4: The pre-registration and reliability analysis of the raters is very good and shows the scientific rigor of the researchers responsible for the study. One thing that could be added is how this type of analysis has been carried out in the field of addiction would be interesting and also what types of benchmarks thy have used.

Response: Thank you for these positive comments regarding the scientific rigor of the present study. We assume the Reviewer #4 is referring to the z-curve analysis with this comment. To our knowledge, z-curve has not been used in the addiction sciences yet. Instead, we compared the results of our results to the results Bartos and Schimmack (2020) obtained using the Open Science Collaboration (2015) dataset. This is a relevant comparison because many researchers in the product safety literature use similar study designs, follow similar reporting conventions, and/or received their graduate training in psychology departments. 

Reviewer 4: The authors have not included a reference or a description of the tetrachoric correlations and why it was in the Methods section. Information about Fisher’s exact test is also missing.

Response: We have clarified in the results section that the use of exact test was unplanned and exploratoryWe replaced the tetrachoric correlations with chi-square tests, as we realized that the assumption underlying tetrachoric correlations-- that the binary variables being correlated are proxies for continuous variables-- does not apply here. We have added a justification for the use of Fisher’s exact test:

“We conducted unplanned, exploratory chi-square tests to explore whether the choice to measure gambling participation or gambling-related problems might have influenced the measurement method. There were significantly more studies of gambling-related problems that used self-reports than would be expected by chance, χ²(1) = 49.10, p < .001, φc = .48. There was a non-significant tendency for studies of gambling participation to not use self-report measures, χ²(1) = 0.52, p = .469, φc = -.08. The non-significance of this pattern of results held when using a Fisher’s exact test (40) to account for expected cell counts with five or fewer cases. A significantly higher proportion of gambling participation studies used gambling records, χ²(1) = 5.98, p = .014, φc = .19, and we obtained the same result using a Fisher’s exact test. Finally, fewer studies of gambling-related problems used gambling records than would be expected by chance, χ²(1) = 4.70, p = .031, φc = -.16.”

Reviewer 4: The results are presented in a clear way. It is long and perhaps it is possible to make it a bit shorter.

Response: We prefer to keep the manuscript length as it is to preserve readability and comprehensiveness in reporting and synthesizing the results.

Reviewer 4: The discussion about should be elaborated using Ladouceur, Shaffer, Blaszczynski, & Shaffer (2019). Responsible gambling research and industry funding biases. Journal of gambling studies, 35(2), 725-730. It can also be discussed in relation to transparency a bit more. What are the potential consequences of this lack of transparency?

Response: This is a good point regarding this cited paper. We now refer to Ladouceur et al. in the Replicability and Transparency section of the discussion:

“Although extant evidence does not support the contention that industry influences the methodology of responsible gambling research (76) , pre-registration would be a worthwhile additional step to ensure independence between researchers and industry actors who fund them (77).”

Reviewer #5: I am not sure that the term "gambler" is appropriate for describing all peoples that gamble, as it has negative connotations and may not reflect the self perception on many who gamble from time to time. The term "player" is less value laden.

Response: This is a fair point to raise. However, we would dispute that ‘gambler’ can be presumed to necessarily have the same or similar negative or stigmatizing connotations that some researchers believe are attached to more specific terms such “problem gambler” or “pathological gambler” (e.g., see https://www.tandfonline.com/doi/full/10.1080/14459795.2020.1808774). Moreover, we are not aware of empirical evidence that using the less precise term ‘player’ has a positive (or negative) effect on regard for people who gamble relative to the term ‘gambler.’ For instance, it is possible that deliberately avoiding using ‘gambler’ tacitly patronizes or condemns individuals who would use the term as a badge of pride or term of endearment. However, due to potential stigmatizing language in the use of “problem gambler” raised in the citation above, in the revised manuscript, we changed all references to “problem gamblers” as “people with gambling-related problems” and “problem gambling” to “gambling-related problems.”

Reviewer 5: Also, I believe the EGM ban in Norway was in July 2007 not 2006.

Response: Thank you for pointing this out. We have changed “2006” to “2007” in the manuscript.

---

## [Decision Letter · Decision Letter 1]

9 Mar 2021

PONE-D-20-27790R1

Responsible Product Design to Mitigate Excessive Gambling: A Scoping Review

PLOS ONE

Dear Dr. McAuliffe,

Thank you for submitting your manuscript to PLOS ONE. After careful consideration, we feel that it has merit but does not fully meet PLOS ONE’s publication criteria as it currently stands. Therefore, we invite you to submit a revised version of the manuscript that addresses the points raised during the review process.

Thank you for taking the time to carefully respond to the reviewers remarks. My reading of the reviews and your response is that we are close to meeting expectations with only one set of comments left to address in this round. If you would respond to these comments I would appreciate that a lot.

We look forward to receiving your revised manuscript.

Kind regards,

Simone N. Rodda

Academic Editor

PLOS ONE

Journal Requirements:

Reviewers' comments:

Reviewer's Responses to Questions

**Comments to the Author**

1. If the authors have adequately addressed your comments raised in a previous round of review and you feel that this manuscript is now acceptable for publication, you may indicate that here to bypass the “Comments to the Author” section, enter your conflict of interest statement in the “Confidential to Editor” section, and submit your "Accept" recommendation.

Reviewer #2: (No Response)

Reviewer #3: (No Response)

Reviewer #4: All comments have been addressed

Reviewer #5: All comments have been addressed

2. Is the manuscript technically sound, and do the data support the conclusions?

Reviewer #2: Yes

Reviewer #3: Partly

Reviewer #4: Yes

Reviewer #5: Yes

3. Has the statistical analysis been performed appropriately and rigorously? 

Reviewer #2: Yes

Reviewer #3: Yes

Reviewer #4: Yes

Reviewer #5: Yes

4. Have the authors made all data underlying the findings in their manuscript fully available?

Reviewer #2: Yes

Reviewer #3: Yes

Reviewer #4: Yes

Reviewer #5: Yes

5. Is the manuscript presented in an intelligible fashion and written in standard English?

Reviewer #2: Yes

Reviewer #3: Yes

Reviewer #4: Yes

Reviewer #5: Yes

6. Review Comments to the Author

Reviewer #2: I appreciate the table but all papers in the table are not as requested in the reference list at the end of the paper.

Reviewer #3: Thanks to the authors for addressing my comments, and those of the other reviewers, in their revision. I’m guessing that they may have been expecting some pushback on some of their comments, so my apologies in advance, but hopefully this is useful feedback and we’re all still friends at the end of it.

In my initial review, I suggested that the authors might consider splitting the paper into two. It seems in their reply that they have indicated that they feel there is not enough on efficacy to be a full paper, and instead their focus is on the replicability and transparency side of things. There seem to be plenty of pages of results about efficacy and maybe three pages about replicability, including a table. In fact, in their reply they say that they are not actually that interested in efficacy per se. The title of the paper has nothing about replicability or transparency, so this isn’t clear. The abstract (which has been rewritten) does seem to cover both sides of things, but I feel that if I went into this study to look for evidence of efficacy, then I’d walk out of it feeling like I’d been deceived into reading about replications instead. Don’t get me wrong, I feel strongly about replication too – it’s crucial to this whole science thing that we’re all trying our best to do. But I’m not sure that I agree with the authors that replication cannot be separated from efficacy for the purposes of a paper, because then you can say that replication cannot be separated from anything and every single paper on anything from now on is going to have to do these analyses. I agree with Reviewer 1, who noted that a lot of this around replication could be an editorial. But I suspect the authors will again rebut this point, and we could go back and forth on this for awhile. I think this will be a call for the editor, and I am of course happy to defer to her views.

The authors have otherwise mostly addressed my concerns, although there are a couple of things that I’ve suggested could be improved throughout, but haven’t been. For example, I pointed out that some studies might have reported BOTH unstandardised and standardised effect sizes, but that other similar things might have been incompletely reported. The authors only addressed the effect sizes example that I highlight. For example, game-based tool or interaction type seems to have some overlap, as percentages add up to more than 100%. There may be other such cases in the paper. As the authors are more familiar with the data, they will be able to identify more of these.

I think the distinction between quasi-experimental and true experimental is actually pretty important, and should be reflected in the new table.

I am not sure that I conveyed my concern around the comparison of the Observed and Estimated Discovery Rates. Both of these things have a point estimate, but also a confidence interval around that point estimate. Just because the ODR point estimate is outside of the EDR confidence interval doesn’t mean statistically significant difference, because the ODR has its own confidence interval and uncertainty which is not taken into account through only using the point estimate of the ODR. The authors, in their revision, have stated that the Observed Discovery Rate is only a point estimate, and that the confidence interval has been omitted, because the sample represents the entire population of interest. I have strong concerns about this, because despite their rigour, there may have been some that were missed for a variety of reasons, or because of errors during search and/or classification, and so on. So, I have a concern with this approach.

To emphasise my point, consider these (obviously made up) data for two groups:

Group 1: 3, 4, 4, 4, 4, 5, 4, 6, 4, 3

Group 2: 6, 3.5, 5, 6, 4, 4, 4, 5, 5, 6

Mean for group 1 is 4.1 (95% CI: 3.47 to 4.73)

Mean for group 2 is 4.85 (95% CI: 4.17 to 5.53)

So the estimate for each is not within the CI of the other.

If I forget that group 2 has a CI, and just compare 4.85 to the CI for group 1, I would conclude that there is a statistically significant difference.

But there is a CI for group 2, and if I run an independent samples t-test (Welch or standard), the p-value is .082, so we would conclude no significant difference. Admittedly, it’s close, but if we’re to use an alpha criterion of .05 for p-values (as absolutely no one ever intended them to be used, and that’s a discussion for another day), it’s not the same as just comparing a single point estimate against a confidence interval, which is what is done in the paper. Given that this is used as evidence for a talking point by the authors, this is pretty important. If this is how the author of the z-curve suggests that the tool is used, then I have hesitations about the procedure, and this harks back to my concerns from my initial review about using new techniques.

Once again, I hope that this discussion is useful. I feel that the authors will probably continue to push back on these points, as is their prerogative, but I do hope that they will find the comments at least vaguely useful.

Reviewer #4: (No Response)

Reviewer #5: (No Response)

7. PLOS authors have the option to publish the peer review history of their article (what does this mean?). If published, this will include your full peer review and any attached files.

Reviewer #2: No

Reviewer #3: **Yes: **Alex M T Russell

Reviewer #4: No

Reviewer #5: No

---

## [Author Response · Author response to Decision Letter 1]

19 Mar 2021

Dear Dr. Rodda,

Thank you for the opportunity to submit a second revised version of our manuscript (which now has a slightly longer title), “Responsible Product Design to Mitigate Excessive Design: A Scoping Review and Z-Curve Analysis of Replicability.” Our responses to the remaining reviewer comments are shown below our signature. We thank the reviewers for the thoughtfulness of their comments, which we believe have strengthened the rigor of our manuscript. We hope that this second revision merits publication in PLOS One.

Sincerely,

William H. B. McAuliffe, Timothy C. Edson, Eric R. Louderback, Alexander LaRaja, and Debi A. LaPlante

Reviewer #2: I appreciate the table but all papers in the table are not as requested in the reference list at the end of the paper.

Response: Thank you for raising this point about the revised table and reference list. We have cross-checked the table with the reference list and added any missing papers to the reference list.

Reviewer #3: In my initial review, I suggested that the authors might consider splitting the paper into two. It seems in their reply that they have indicated that they feel there is not enough on efficacy to be a full paper, and instead their focus is on the replicability and transparency side of things. There seem to be plenty of pages of results about efficacy and maybe three pages about replicability, including a table. In fact, in their reply they say that they are not actually that interested in efficacy per se. The title of the paper has nothing about replicability or transparency, so this isn’t clear. The abstract (which has been rewritten) does seem to cover both sides of things, but I feel that if I went into this study to look for evidence of efficacy, then I’d walk out of it feeling like I’d been deceived into reading about replications instead. Don’t get me wrong, I feel strongly about replication too – it’s crucial to this whole science thing that we’re all trying our best to do. But I’m not sure that I agree with the authors that replication cannot be separated from efficacy for the purposes of a paper, because then you can say that replication cannot be separated from anything and every single paper on anything from now on is going to have to do these analyses. I agree with Reviewer 1, who noted that a lot of this around replication could be an editorial. But I suspect the authors will again rebut this point, and we could go back and forth on this for awhile. I think this will be a call for the editor, and I am of course happy to defer to her views.

Response: We do appreciate the reviewer's perspective about this issue. However, we remain convinced that keeping the paper as a whole provides a new and interesting perspective about this literature. As we mention in the introduction, other review studies are available, yet these papers do not examine the replicability of this research literature. But we do not wish to mislead readers about the focus of our study, so we have revised our title to “Responsible Product Design to Mitigate Excessive Gambling: A Scoping Review and Z-Curve Analysis of Replicability.” We hope that this revised title better sets readers' expectations and emphasizes our position that replicability is relevant to every discussion of empirical data, and the absence of such discussion might lead readers to make unwarranted assumptions about efficacy. One final note, we would make the same points with regards to the relationship between efficacy and methodological rigor. We did not focus on methodological rigor in the present paper, because (as we noted in the paper) previous reviews have discussed methodological limitations in detail and the state of play has not much changed since then.

Reviewer #3:The authors have otherwise mostly addressed my concerns, although there are a couple of things that I’ve suggested could be improved throughout, but haven’t been. For example, I pointed out that some studies might have reported BOTH unstandardised and standardised effect sizes, but that other similar things might have been incompletely reported. The authors only addressed the effect sizes example that I highlight. For example, game-based tool or interaction type seems to have some overlap, as percentages add up to more than 100%. There may be other such cases in the paper. As the authors are more familiar with the data, they will be able to identify more of these.

I think the distinction between quasi-experimental and true experimental is actually pretty important, and should be reflected in the new table.

Response: Thank you for these important points about items reported from each study. We agree with this suggestion and now indicate in Table 2 when a study was quasi-experimental. We also do note in Table 2 when a paper tested more than one type of tool. Also, we would like to kindly remind the reviewer that the first sentence of the “Game-based Tool or Interaction Type” section of the results is “Some studies (n = 7) tested multiple types of responsible product designs.”

Reviewer #3: I am not sure that I conveyed my concern around the comparison of the Observed and Estimated Discovery Rates. Both of these things have a point estimate, but also a confidence interval around that point estimate. Just because the ODR point estimate is outside of the EDR confidence interval doesn’t mean statistically significant difference, because the ODR has its own confidence interval and uncertainty which is not taken into account through only using the point estimate of the ODR. The authors, in their revision, have stated that the Observed Discovery Rate is only a point estimate, and that the confidence interval has been omitted, because the sample represents the entire population of interest. I have strong concerns about this, because despite their rigour, there may have been some that were missed for a variety of reasons, or because of errors during search and/or classification, and so on. So, I have a concern with this approach.

To emphasise my point, consider these (obviously made up) data for two groups:

Group 1: 3, 4, 4, 4, 4, 5, 4, 6, 4, 3

Group 2: 6, 3.5, 5, 6, 4, 4, 4, 5, 5, 6

Mean for group 1 is 4.1 (95% CI: 3.47 to 4.73)

Mean for group 2 is 4.85 (95% CI: 4.17 to 5.53)

So the estimate for each is not within the CI of the other.

If I forget that group 2 has a CI, and just compare 4.85 to the CI for group 1, I would conclude that there is a statistically significant difference.

But there is a CI for group 2, and if I run an independent samples t-test (Welch or standard), the p-value is .082, so we would conclude no significant difference. Admittedly, it’s close, but if we’re to use an alpha criterion of .05 for p-values (as absolutely no one ever intended them to be used, and that’s a discussion for another day), it’s not the same as just comparing a single point estimate against a confidence interval, which is what is done in the paper. Given that this is used as evidence for a talking point by the authors, this is pretty important. If this is how the author of the z-curve suggests that the tool is used, then I have hesitations about the procedure, and this harks back to my concerns from my initial review about using new techniques.

Response: We now note in the limitations section that accidentally omitting relevant studies “might have affected the conclusions we drew about replicability and publication bias based on the results from z-curve.” That said, after reflection on your comment, we do not think that omitting the confidence interval of the Observed Discovery Rate is one of the potential weaknesses of z-curve. Although it is true that we might have overlooked some hard-to-find studies, we do not believe that the confidence interval would adequately represent this source of uncertainty. Rather, the confidence interval represents uncertainty from having analyzed a random sample from a larger population. For instance, reporting the confidence interval around the Observed Discovery Rate would have been appropriate had we decided that coding the p-values from all studies was too much work, and so we deliberately only coded a random subsample of them. The source of uncertainty presently in question is not random—there would have to be something about the study(ies) that made it (them) hard to find or difficult to correctly classify as relevant, which might in turn be related to its (their) replicability.

---

## [Editor Report · Decision Letter 2]

29 Mar 2021

Responsible Product Design to Mitigate Excessive Gambling: A Scoping Review and Z-Curve Analysis of Replicability

PONE-D-20-27790R2

Dear Dr. McAuliffe,

We’re pleased to inform you that your manuscript has been judged scientifically suitable for publication and will be formally accepted for publication once it meets all outstanding technical requirements.

On a personal note - thank you for responding to the reviewer comments so clearly across the various rounds.

Kind regards,

Simone N. Rodda

Academic Editor

PLOS ONE
---

## [Editor Report · Acceptance letter]

8 Apr 2021

PONE-D-20-27790R2 

Responsible Product Design to Mitigate Excessive Gambling: A Scoping Review and Z-Curve Analysis of Replicability 

Dear Dr. McAuliffe:

I'm pleased to inform you that your manuscript has been deemed suitable for publication in PLOS ONE. Congratulations! Your manuscript is now with our production department. 

Kind regards, 

on behalf of

Dr. Simone N. Rodda 

Academic Editor

PLOS ONE